# Delphic Offline Reinforcement Learning under Nonidentifiable Hidden Confounding

**Alizée Pace** [1,2,3]        **Hugo Yèche** [2]

**Bernhard Schölkopf** [3]        **Gunnar Rätsch** [1,2]        **Guy Tennenholtz** [4]

[1] ETH AI Center        [2] Department of Computer Science, ETH Zürich
[3] Max Planck Institute for Intelligent Systems, Tübingen
[4] Google Research
`alizee.pace@ai.ethz.ch`

## Abstract

A prominent challenge of offline reinforcement learning (RL) is the issue of hidden confounding: unobserved variables may influence both the actions taken by the agent and the observed outcomes. Hidden confounding can compromise the validity of any causal conclusion drawn from data and presents a major obstacle to effective offline RL. In the present paper, we tackle the problem of hidden confounding in the nonidentifiable setting. We propose a definition of uncertainty due to hidden confounding bias, termed delphic uncertainty, which uses variation over world models compatible with the observations, and differentiate it from the well-known epistemic and aleatoric uncertainties. We derive a practical method for estimating the three types of uncertainties, and construct a pessimistic offline RL algorithm to account for them. Our method does not assume identifiability of the unobserved confounders, and attempts to reduce the amount of confounding bias. We demonstrate through extensive experiments and ablations the efficacy of our approach on a sepsis management benchmark, as well as on electronic health records. Our results suggest that nonidentifiable hidden confounding bias can be mitigated to improve offline RL solutions in practice.

## 1 Introduction

Large observational datasets for decision-making open the possibility of learning expert policies with minimal environment interaction. This holds promise for contexts where exploration is impractical, unethical or even impossible, such as optimising marketing, educational or clinical decisions based on relevant historical datasets (Gottesman et al., 2018; Singla et al., 2021; Thomas et al., 2017). Recent years have thus seen the emergence of offline reinforcement learning (RL) literature (Levine et al., 2020), which proposes to adapt RL methods to overcome estimation biases induced by learning from finite, fully offline data.

Aside from estimation biases, confounding variables are common in offline data (Gottesman et al., 2018). The problem of hidden confounding, where outcome and decisions are both dependent on an unobserved factor, is widely overlooked in many of the concurrent offline RL methods. Nevertheless, it may induce significant errors, even for the simplest of bandit problems, and is especially aggravated in the sequential setting (Chakraborty and Murphy, 2014; Tennenholtz et al., 2022; Zhang and Bareinboim, 2019). Hidden confounding exists in numerous applications. In autonomous driving, for example, the observational policy may behave according to unobserved factors (e.g. road conditions (Haan et al., 2019)), which also affect environment dynamics and rewards. Alternatively, in the medical context, unrecorded patient state information such as socio-economic factors or visual appearance may have been taken into account by the acting physician (Gottesman et al., 2018).

In this work, we focus on *nonidentifiable* hidden confounding in offline RL. While prior work has mostly addressed the problem in the identifiable setup (Kumor et al., 2021; Lu et al., 2018a; Wang et al., 2021; Zhang and Bareinboim, 2020), we show that significant improvement in policy learning can be achieved even in the realistic nonidentifiable setting. We propose an approach to estimate uncertainty due to confounding bias and to account for the degree of confoundedness while learning. In turn, this leads to improved downstream performance for offline learning algorithms.

Our main contributions are as follows. (1) To the best of our knowledge, we are the first to address *nonidentifiable* confounding bias in *deep offline RL*. (2) We achieve this by introducing a novel uncertainty quantification method from observational data, which we term delphic uncertainty. (3) We propose an offline RL algorithm that leverages this uncertainty to obtain confounding-averse policies, and (4) we demonstrate its performance on both synthetic and real-world medical data.

## 2 PRELIMINARIES

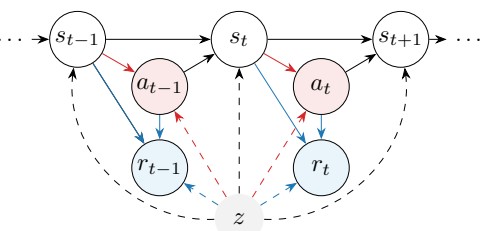

We consider the contextual Markov Decision Process (MDP) (Hallak et al., 2015), defined by the tuple $\mathcal{M} = (\mathcal{S}, \mathcal{Z}, \mathcal{A}, T, r, \rho_0, \nu, \gamma)$, where $\mathcal{S}$ is the state space, $\mathcal{A}$ is the action space, $\mathcal{Z}$ is the context space, $T : \mathcal{S} \times \mathcal{Z} \times \mathcal{A} \rightarrow \Delta\mathcal{S}$ is the transition function, $r : \mathcal{S} \times \mathcal{Z} \times \mathcal{A} \rightarrow [0, 1]$ is the reward function, and $\gamma \in [0, 1)$ is the discount factor. We assume an initial state distribution $\rho_0 : \mathcal{Z} \rightarrow \Delta\mathcal{S}$ and a context distribution $\nu$, such that each inter-action episode has a fixed context $z \sim \nu$, and the environment initialises at state $s_0 \sim \rho_0(\cdot|z)$. At time $t$, the environment is at state $s_t \in \mathcal{S}$ and an

Figure 1: **Contextual MDP.** Black arrows show the transition dynamics, blues ones the reward function, and red ones the policy. Confounding arises when both behavioural policy $\pi_b$ and environment returns depend on hidden context variable $z$ (dashed lines).

agent selects an action $a_t \in \mathcal{A}$. The agents receives a reward $r_t = r(s_t, a_t, z)$ and the environment then transitions to state $s_{t+1} \sim T(\cdot|s_t, a_t, z)$. A causal graph of the process is depicted in Figure 1.

We define a *context-aware* policy $\pi$ as a mapping $\pi : \mathcal{S} \times \mathcal{Z} \rightarrow \Delta\mathcal{A}$, such that $\pi(a|s, z)$ is the probability of taking action $a$ in state $s$ and context $z$. Likewise, we define a *context-independent* policy as $\tilde{\pi} : \mathcal{S} \rightarrow \Delta\mathcal{A}$. We denote the set of all such policies as $\Pi$ and $\tilde{\Pi}$, respectively.[1]

We assume access to a dataset of $N$ trajectories $\mathcal{D} = \{\tau^i\}_{i=1}^N$, where the sequences $\tau^i = (s_0^i, a_0^i, r_0^i, \ldots s_H^i, a_H^i, r_H^i)$ are trajectories induced by an unknown, context-aware behavioural policy $\pi_b \in \Pi$ such that $a_t^i \sim \pi_b(\cdot|s_t^i, z^i)$. The decision-making context $z^i$ for each trajectory is *not* included in the observational dataset. In the following, we drop index $i$ unless explicitly needed.

Finally, we define the offline RL task, which consists of finding an optimal context-independent policy $\tilde{\pi}^* \in \tilde{\Pi}$ – one which maximises the expected discounted returns. Specifically, we define the state-action value function of a policy $\tilde{\pi} \in \tilde{\Pi}$ by $Q^{\tilde{\pi}}(s, a) = \mathbb{E}_{\tilde{\pi}, z \sim \nu}[\sum_{t=0}^\infty \gamma^t r(s_t, a_t, z)|s_0 = s, a_0 = a]$, where $\mathbb{E}_{\tilde{\pi}}$ denotes the expectation induced by following policy $\tilde{\pi} \in \tilde{\Pi}$. We also define the value of $\tilde{\pi}$ by $V^{\tilde{\pi}}(s) = \mathbb{E}_{a \sim \tilde{\pi}}[Q^{\tilde{\pi}}(s, a)]$. An optimal policy is then defined by $\tilde{\pi}^*(\cdot|s) = \arg\max_{\tilde{\pi} \in \tilde{\Pi}}[V^{\tilde{\pi}}(s)]$.

## 3 SOURCES OF ERROR IN OFFLINE RL

Optimising a policy from observational data is prone to various sources of error, which many RL works propose to decompose, estimate, and bound (Levine et al., 2020; Tennenholtz and Mannor, 2022). First, the process is prone to statistical error in correctly estimating a value model from the observed data (Jin et al., 2021). Inherent environment stochasticity (aleatoric uncertainty) can result in imprecise models, whereas finite data quantities (epistemic uncertainty) can lead to poor model approximation. Improper handling of estimation errors causes covariate shift and overestimation problems, as evident in behaviour cloning (Ross et al., 2011) and offline RL (Kumar et al., 2020). Offline RL approaches typically mitigate these errors through pessimism, penalising areas where error is expected to be large (Jin et al., 2021; Levine et al., 2020).

Another source of error, which is often overlooked in the RL literature, is structural bias. Independent of data quantity, such a bias can occur when the state-action space coverage is incomplete (Uehara and Sun, 2022), or when the expressivity of the model class considered is inappropriate (Lu et al., 2018b). Our work considers confounding bias – a critical type of structural bias, which is often disregarded despite many data collection environments being prone to its occurrence (Gottesman et al., 2018; Haan et al., 2019; Kallus and Zhou, 2018; 2020).

---

[1]One can also consider history-dependent policies. Nevertheless, Markov policies sufficiently illustrate the challenges of our task, and can be easily generalised to history-dependent ones.

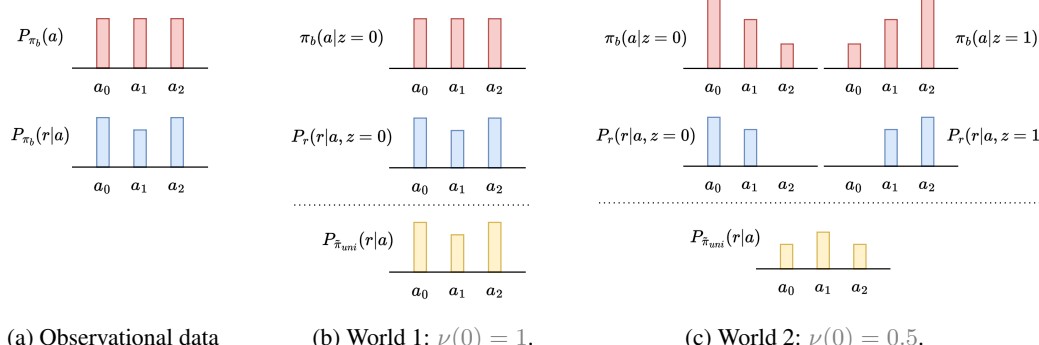

(a) Observational data      (b) World 1: $\nu(0) = 1$.      (c) World 2: $\nu(0) = 0.5$.

Figure 2: **Confounding Bias Example.** World 1 and 2 are two models for the binary confounding variable that are compatible with the marginalised observational data in (a), composing models for $\nu(z)$, $\pi_b(a|z)$ and $P(r|a, z)$. Under an alternative policy, such as a context-independent uniform policy $\tilde{\pi}_{uni}(\cdot) = 1/|\mathcal{A}|$, these two worlds give different values to each action.

**Confounding Bias.** Confounding bias arises when the observational policy depends on unobserved factors which also affect the chosen action and the reward or transition function (Tennenholtz et al., 2022). To better understand how confounding bias may affect offline RL algorithms, consider the process detailed in Section 2 and depicted in Figure 1. The offline data was generated by sampling trajectories from the behavioural policy distribution $\tau \sim P_{\pi_b}(\tau)$, which is marginalised over $\nu(z)$ and factorises as follows:

$$P_{\pi_b}(\tau) = \mathbb{E}_{z \sim \nu}\left[\rho_0(s_0|z) \prod_{t=0}^{H} \pi_b(a_t|s_t, z)\, P_r(r_t|s_t, a_t, z)\, T(s_{t+1}|s_t, a_t, z)\right], \quad (1)$$

where $P_r$ is the probability of sampling reward $r_t$ from $r(s_t, a_t, z)$. Any offline reinforcement learning objective can be written as an expectation over this trajectory distribution (Levine et al., 2020). Confounding arises when one learns models on trajectories following $P_{\pi_b}(\tau)$, but estimates the value of policies $\pi$ that *change* the probability of taking an action $a$ in a given state and context $(s, z)$ – as is necessarily the case when considering context-independent policies. Since all model terms in Equation (1) are unknown and nonidentifiable due to their dependence on $z$, there may exist several "worlds" that could induce the same observational distribution $P_{\pi_b}(\tau)$. This is known as the *identifiability problem* in the causal inference literature (Kallus and Zhou, 2018; Namkoong et al., 2020), which has been studied extensively, providing methods for analyzing when counterfactual estimates can be obtained. Particularly, without additional assumptions about the causal structure of the environment – such as using environment interventions (Lu et al., 2018a; Zhang and Bareinboim, 2020) or the existence of observable back- or front-door variables (Kumor et al., 2021; Wang et al., 2021) – the context $z$ acting as confounder is nonidentifiable and cannot be estimated. Below, we illustrate through a simple example, how two equally plausible models can correctly construct the same observational distribution, yet induce two different values for another policy.

**An Illustrative Example.** Suppose access to the bandit data in Figure 2a, induced by an unknown context-aware policy $\pi_b$ with marginal distribution $P_{\pi_b}(a, r)$. Simplifying Equation (1) to this setup, we obtain:

$$P_{\pi_b}(a, r) = \mathbb{E}_{z \sim \nu}[\pi_b(a|z) P_r(r|a, z)].$$

We can therefore change $\nu$, $\pi_b$, and $P_r$ to induce the same marginalised distribution $P_{\pi_b}$, with a significant difference in reward for a counterfactual policy. Indeed, in Figure 2b, World 1 assumes a singleton context with a uniform behavioural policy, whereas World 2 (Figure 2c) assumes two contexts and a behavioural policy which changes its distribution w.r.t. the sampled context. In both of these worlds, the observational distribution $P_{\pi_b}(a, r)$ remains the same, whereas the uniform policy $\tilde{\pi}_{uni}(\cdot) = 1/|\mathcal{A}|$ results in different reward distributions and optimal actions. Without explicit access to the ground-truth context, modelling an alternative policy to the privileged data-generating one will therefore be prone to spurious correlations and estimation biases.

In the next section, we propose to address the general *nonidentifiable* confounding problem in offline RL by estimating the amount of possible confounding bias within the observational dataset and correcting for it during learning. We compare our approach to related work in Table 1 and in more detail in Section 7.

Table 1: **Comparison of related work addressing confounding bias in decision-making**, further discussed in Section 7. Our formalism can be built upon with different parametric assumptions. For instance, the sensitivity analysis assumption could be applied to restrict the set of the compatible worlds considered in delphic uncertainty. The identifiable proxy variable case results in a singleton set of compatible worlds , $|\mathcal{W}| = 1$.

| Related work | Modelling assumptions | Pessimism strategy | Offline RL algorithm |
|---|---|---|---|
| Offline RL (Levine et al., 2020) | No confounding | Penalised $Q^\pi$ | ✓ |
| Environment interaction (Zhang and Bareinboim, 2020) | $P_\pi(\tau)$ accessible | ✗ | ✗ |
| Proxy variables (Wang et al., 2021) | $\nu(z)$ identifiable | ✗ | ✓ |
| Sensitivity analysis (Kallus and Zhou, 2020) | $\pi_b(a\|s,z)/\pi_b(a\|s,z') < \Gamma$ | Worst-case $Q^\pi$ | ✗ |
| Delphic offline RL (**Ours**) | Set $\mathcal{W}$ of possible $\nu(z)$ | Penalised $Q^\pi$ | ✓ |

# 4    MEASURING CONFOUNDING BIAS THROUGH DELPHIC UNCERTAINTY

In this section, we formulate a method for estimating uncertainty arising from confounding bias in offline RL, which we term *delphic* uncertainty[2]. While aleatoric and epistemic uncertainty can be expressed as probability distributions over model outputs and parameters, respectively (Hüllermeier and Waegeman, 2021), delphic uncertainty is a distribution over counterfactual values. We propose a general approach to decouple aleatoric, epistemic, and delphic uncertainties, which we leverage to overcome confounding bias in Section 5.

We first formalise the notion of world models compatible with the observational distribution $P_{\pi_b}(\tau)$:

**Definition 4.1.** A compatible world for $P_{\pi_b}$ is a tuple $w = (\mathcal{Z}_w, \nu_w, \rho_{0,w}, P_{r,w}, T_w, \pi_{b,w})$ which satisfies $P_{\pi_b}(\tau) = \mathbb{E}_{z \sim \nu_w}\left[\rho_{0,w}(s_0|z) \prod_{t=0}^{H} \pi_{b,w}(a_t|s_t,z)P_{r,w}(r_t|s_t,a_t,z)T_w(s_{t+1}|s_t,a_t,z)\right]$ for any trajectory $\tau$. We denote by $\mathcal{W}$ the set of all compatible worlds.

We focus on uncertainty estimates of value functions. Let $w \in \mathcal{W}$ (i.e., $w$ is some compatible world for $P_{\pi_b}$). We use $\theta_w$ to denote the parameters of a Q-value function in world $w$. For a fixed $w$ and $\theta_w$, we assume each value model $Q_{\theta_w}$ is defined by some stochastic model, e.g., a normal distribution $Q_{\theta_w}|\theta_w, w \sim \mathcal{N}\left(\mu_{\theta_w}, \sigma_{\theta_w}^2\right)$. Indeed, here $\sigma_{\theta_w}$ accounts for **aleatoric uncertainty**, capturing the intrinsic stochasticity of the environment (Kendall and Gal, 2017). Additional statistical uncertainty arises from the distribution over model parameters $\theta_w$ in the fixed world $w \in \mathcal{W}$. Starting from a prior over $\theta_w$, evidence from the data leads to a posterior estimate over the correct model parameters $P(\theta_w|w)$, which captures **epistemic uncertainty** (Hüllermeier and Waegeman, 2021). We refer the interested reader to Appendix A.1 for an overview of statistical uncertainty estimation methods.

We are now ready to define the uncertainty induced by confounding variables, which we term **delphic uncertainty**. To do this, we leverage Definition 4.1 and define delphic uncertainty by varying over compatible world models. Following on prior work separating epistemic and aleatoric uncertainty (Kendall and Gal, 2017), we can decompose the variance in the value function estimate between the three types of uncertainties. Particularly, let $w$ be a compatible world for $P_{\pi_b}$, and let $P(w)$ be some distribution over worlds in $\mathcal{W}$. This posits the following distribution for $Q_{\theta_w}$: $P(Q_{\theta_w}) = \int \int P(Q_{\theta_w}|\theta_w, w)P(\theta_w|w)P(w)d\theta_w dw$. We have the following result. Its proof, based on the law of total variance (Weiss et al., 2006), is given in Appendix B.

**Theorem 4.2** (Variance Decomposition). *For any $\pi \in \Pi$, we have*

$$\text{Var}(Q_{\theta_w}^\pi) = \mathbb{E}_w\left[\underbrace{\mathbb{E}_{\theta_w}\left[\text{Var}(Q_{\theta_w}^\pi|\theta_w, w)|w\right]}_{aleatoric\ uncertainty} + \underbrace{\text{Var}_{\theta_w}\left(\mathbb{E}[Q_{\theta_w}^\pi|\theta_w, w]|w\right)}_{epistemic\ uncertainty}\right] + \underbrace{\text{Var}_w(\mathbb{E}_{\theta_w}[\mathbb{E}[Q_{\theta_w}^\pi|\theta_w, w]|w])}_{delphic\ uncertainty}.$$

To gain further intuition of this result, consider the case of normal distributions. We can rewrite Theorem 4.2 as:

$$\text{Var}(Q_{\theta_w}^\pi) = \mathbb{E}_w\left[\mathbb{E}_{\theta_w}[\sigma_{\theta_w}|w]^2 + \text{Var}_{\theta_w}(\mu_{\theta_w}|w) + \text{Var}_{\theta_w}(\sigma_{\theta_w}|w)\right] + \text{Var}_w(\mathbb{E}_{\theta_w}[\mu_{\theta_w}|w]) \quad (2)$$

The first three terms, calculated through statistics of predicted means and standard deviations, capture aleatoric and epistemic uncertainties, whereas the final term, calculated through variation over compatible world models, captures delphic uncertainty. Indeed, the latter form of uncertainty remains even in deterministic environments ($\sigma_{\theta_w} \to 0$, no aleatoric uncertainty) and infinite data ($|\mathcal{D}| \to \infty$, no epistemic uncertainty). We refer the reader to Appendix B.2 for further discussion.

---

[2]The word "delphic" characterises quantities that are ambiguous and opaque, relating to the hidden confounding variables and their elusive effect on model predictions.

# 5 OFFLINE RL UNDER DELPHIC UNCERTAINTY

In the previous section, we defined delphic uncertainty through variation over compatible world models. We now propose a method to measure it in practice, then leverage our uncertainty estimate within an offline reinforcement learning framework to mitigate confounding bias.

## 5.1 MEASURING DELPHIC UNCERTAINTY

Following the estimation approach outlined in Theorem 4.2, delphic uncertainty can be measured through the variation over value functions $Q^\pi$ from different world models $w$. We propose one method to find a set of world models; note that alternative partial identification assumptions (Kallus and Zhou, 2018; Wang et al., 2021) could be adopted.

**Modelling Compatible Worlds using Variational Inference.** Mirroring the illustrative example in Section 3, a compatible world $w \in \mathcal{W}$ must capture the following key components, using observational data, and grounded by the estimation of the counterfactual action-value function $Q^\pi$:

1. A model for the behavioural policy $\pi_{b,w}(a|s,z)$ – s.t. the world $w$ can be adjusted to $\pi$;

2. A model for the action-value function $Q_w^{\pi_b}(s,a,z)$ under the behavioural policy;

3. A confounder distribution $\nu_w(z)$ within the world $w$.

With these elements, the value function of a policy $\pi$ can be estimated over one step using importance sampling and marginalising over $z$: $Q_w^\pi(s,a) = \mathbb{E}_{z \sim \nu_w(z)}[\pi(a|s)/\pi_{b,w}(a|s,z) \, Q_w^{\pi_b}(s,a,z)]$. We discuss alternative counterfactual estimation approaches in Appendix C.

We propose to train all elements jointly by maximising the likelihood of the first two terms over the behavioural data, marginalising over a latent variable model for $\nu_w(z)$. In fact, starting from a plausible prior, we can learn a data-driven posterior for the confounder distribution $\nu_w(z|\tau)$ which leverages information about $z$ gained over the course of an observed trajectory. This allows the world model to converge to the ground-truth $\nu$ (and to the true dependence of $Q^{\pi_b}$ and $\pi_b$ on $z$) in the identifiable setting.

The compatible world is therefore trained through variational inference, using state-actions $(s,a) \in \tau$ and sampling $z \sim \nu_w(z|\tau)$ through the reparametrisation trick for each trajectory $\tau \in \mathcal{D}$. Optimal parameters are obtained by maximising the Evidence Lower Bound (ELBO, Kingma and Welling (2014)) over $\mathcal{D}$; namely,

$$\mathbb{E}_{(s,a) \in \tau; \, z \sim \nu_w(z|\tau)} \big[ \log \pi_{b,w}(a|s,z) + \alpha \log Q_w^{\pi_b}(s,a,z) \big]$$
$$- \beta D_{KL}\big( \nu_w(z|\tau) \, \| \, p(z) \big),$$

Figure 3: **World model architecture** $w = (\nu, \pi_b, Q^{\pi_b})$, under a prior $p(z)$ for the confounder distribution. Variance in $Q^\pi$ over multiple worlds captures delphic uncertainty.

where $\{\alpha, \beta\}$ are hyperparameters (Higgins et al., 2017), $p(z)$ is a prior over the confounder distribution and $D_{KL}$ is the Kullback-Leibler divergence. Observed action probabilities and Monte-Carlo estimates are used as targets for $\pi_{b,w}$ and $Q_w^{\pi_b}$.

**Counterfactual Variation Across Worlds.** The set of compatible worlds $\mathcal{W}$ for the observational training dataset can be approximated by an ensemble of $W$ world models, each trained using different confounder space dimensionalities $|\mathcal{Z}|$, prior distributions $p(z)$ and model architectures – we propose specific implementation details in Appendix C. Epistemic and aleatoric uncertainty can be separately captured by implementing each world model element as an ensemble of probabilistic models. Following our definition of confounding bias (Theorem 4.2), we measure delphic uncertainty through the variance in $Q_w^\pi(s,a)$ across worlds: $u_d^\pi(s,a) = \text{Var}_w(Q_w^\pi(s,a))$ for policy $\pi$ at state-action $(s,a)$, taking the mean $Q^\pi$ output, averaged over the world model ensemble.

When no confounding exists, all models in $\mathcal{W}$ should identify similar $\nu, Q^{\pi_b}$ and $\pi_b$, returning a similar value of $Q^\pi$. On the other hand, confounding with ambiguous returns would lead to different values across world models. While one could theoretically consider all possible world models in Definition 4.1, we found that, in practice, varying over a subset of compatible models was enough to show improved offline RL efficiency. We refer the reader to Appendix C for an exhaustive overview of the training procedure.

---

**Algorithm 1:** Delphic Offline Reinforcement Learning

---

1 **Input:** *Observational dataset $\mathcal{D}$, Offline RL algorithm.*
2 Learn compatible world models $\{\mathcal{Z}_w, \nu_w, \rho_{0,w}, P_{r,w}, T_w, \pi_{b,w}\}_{w \in \mathcal{W}}$ that all factorise to $P_{\pi_b}(\tau)$.
3 Obtain counterfactual predictions $Q_w^\pi$ for each $w \in \mathcal{W}$.
4 Define local delphic uncertainty: $u_d^\pi(s, a) = \mathrm{Var}_w(Q_w^\pi(s, a))$.
5 Apply pessimism using $u_d$ in Offline RL algorithm (see Section 5.2)

---

## 5.2 Delphic ORL: Offline Reinforcement Learning under Delphic Uncertainty

Inspired by pessimistic approaches in offline RL (Fujimoto et al., 2019; Jin et al., 2021; Kumar et al., 2019; 2020; Levine et al., 2020), we propose to penalise the value of states and actions where delphic uncertainty is high, such that the learned policy is less likely to rely on spurious correlations between actions, states and rewards. This pessimistic approach, which enables the agent to account for and mitigate confounding bias when making decisions, is summarised in Algorithm 1.

In this paper, we incorporate pessimism with respect to delphic uncertainty by modifying the target $Q_{target}$ for the Bellman update in a model-free offline RL algorithm to

$$Q'_{target}(s, a) = Q_{target}(s, a) - \lambda u_d^\pi(s, a),$$

where $\pi$ is the latest learned policy, $(s, a)$ is a tuple sampled for the update and hyperparameter $\lambda$ controls the penalty strength. This approach enjoys the performance guarantees established in prior offline RL work penalising reward of value functions with oracle error functions (Levine et al., 2020; Yu et al., 2020). We apply our penalty to Conservative Q-Learning (Kumar et al., 2020), but could also implement it within any model-free offline RL algorithm – which already induces pessimism with respect to epistemic uncertainty (Fujimoto and Gu, 2021; Kumar et al., 2020; Levine et al., 2020). Similar to epistemic uncertainty penalties in offline RL (Fujimoto and Gu, 2021), we conjecture that ours may act as a form of regularisation towards the observed policy.

Note that various other methods can be adopted to drive pessimism against delphic uncertainty within existing offline RL algorithms (Fujimoto and Gu, 2021; Kumar et al., 2020; Yu et al., 2020), depending on the task at hand. A penalty can be subtracted from the reward function in model-based methods (Yu et al., 2020). The uncertainty measure can also be used to identify a subset of possible actions (Fujimoto et al., 2019), or to weigh samples in the objective function, prioritising unconfounded data. We refer the reader to Appendix C for implementation details.

## 6 Experiments

In this section, we study the benefits of our proposed delphic uncertainty estimation method and its application in offline RL. We validate two principal claims: (1) Our delphic uncertainty measure captures bias due to hidden confounders. (2) Algorithm 1 leads to improved offline RL performance in both simulated and real-world confounded decision-making problems, compared to state-of-the-art but biased approaches. As baselines compatible with the discrete action spaces of environments studied here, we consider Conservative Q-Learning (CQL) (Kumar et al., 2020), Batch-Constrained Q-Learning (BCQ) (Fujimoto et al., 2019) and behaviour cloning (BC) (Bain and Sammut, 1996). Implementation and dataset details are provided in Appendices C and D respectively. In the following, we measure and vary confounding strength through the dependence of the behavioural policy on the hidden confounders, $\Gamma = \max_{z,z' \in \mathcal{Z}, s \in \mathcal{S}, a \in \mathcal{A}}[\pi_b(a|s, z)/\pi_b(a|s, z')]$ (Rosenbaum, 2002), where $z$ also affects the transition dynamics or reward function.

### 6.1 Sepsis Simulation

We explore a simulation of patient evolution in the intensive care unit adapted from Oberst and Sontag (2019). The diabetic status of a patient, accessible to the near-optimal behavioural policy but absent from the observational dataset, acts as hidden confounder $z$.

**Uncertainty Measures.** First, we study the relationship between our uncertainty estimates and the decision-making setup. In Figure 4, we find that epistemic uncertainty reduces with greater data quantities and increases out of the training set distribution, whereas aleatoric uncertainty increases with environment stochasticity, in agreement with prior work (Kendall and Gal, 2017). Our delphic uncertainty estimate, on the other hand, cannot be reduced with more data and increases with greater confounding. Moreover, we found that delphic uncertainty relates to meaningful regions of state-action space, as it is highest under vasopressor administration – the only treatment for which patient

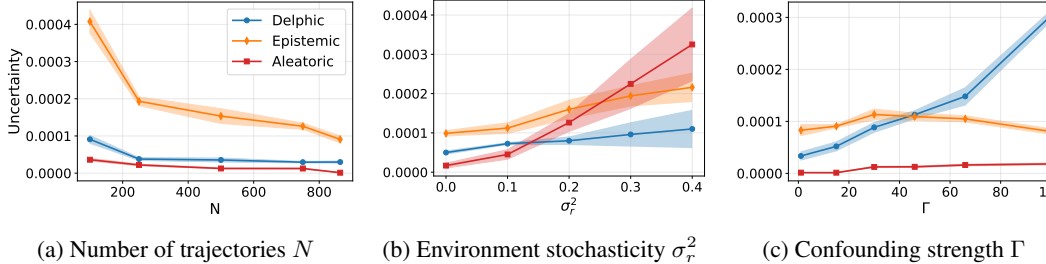

(a) Number of trajectories $N$     (b) Environment stochasticity $\sigma_r^2$     (c) Confounding strength $\Gamma$

Figure 4: **Uncertainty measures as a function of data properties**, averaged over state-action pairs in the sepsis dataset. Epistemic uncertainty reduces most with more data, aleatoric uncertainty increases most with environment stochasticity (reward variance), and delphic uncertainty increases most with confounding strength.

evolution is confounded by the hidden diabetic status, and which is therefore penalised more strongly as confounding strength $\Gamma$ increases. We refer the reader to Appendix E for an exhaustive overview and further experiments.

**Offline RL Performance.** In Figure 5, we compare environment returns obtained through offline RL, imitation learning, and our proposed approach. Our results reveal the susceptibility of offline RL to confounding bias: the presence of unobserved factors $z$ that influence both the behaviour policy and transition dynamics leads to inaccurate value function estimates. Behaviour cloning appears to be less prone to this bias but still faces challenges in dealing with missing information in $z$, evidenced by the performance gap to the online context-aware policy in the unconfounded case ($\Gamma = 1$), and with the distribution shift in observed histories (Ortega et al., 2021). In contrast, our approach to penalising delphic uncer-

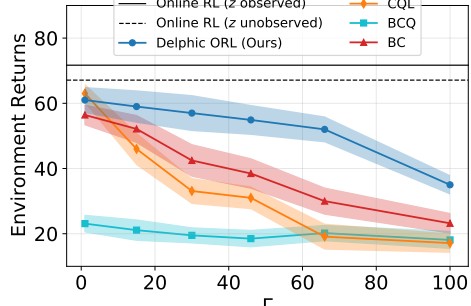

Figure 5: **Performance Results** as a function of confounding strength $\Gamma$. Normalised environment returns (mean and shaded 95% CIs) over 10 runs.

tainty leads to superior performance, especially as confounding strength increases. In Appendix E, we also compare different approaches to implementing pessimism w.r.t. delphic uncertainty, as detailed in Section 5.2, measure the effects of incorporating partial identification assumptions (Kallus and Zhou, 2018) or a dependence on history, and provide an ablation over performance as a function of pessimism hyperparameter $\lambda$.

## 6.2 REAL-WORLD DATA

We demonstrate the added value of our algorithm in optimising decision-making policies from real-world medical data. Our clinical policies are trained using a publicly available dataset of electronic health records, with over 33 thousand patient stays in intensive care and over 200 measured variables HiRID, Hyland et al. (2020)). We consider the problem of optimising the treatment policy for vasopressor and fluid administration[3], and design the reward function to avoid states of circulatory failure. Significant information about patients' conditions is not available in the dataset, despite being critical to the treatment choices of attending physicians, such as socio-economic factors or medical history (Yang and Lok, 2018). For ease of evaluation, we introduce additional, artificial confounders by excluding them from the observational dataset, focusing on diagnostic indicator variables, age and weight ($\Gamma \in [1, 200]$). Disease severity is measured through the SOFA score system (Vincent et al., 1996). Recent RL methods for clinical data could improve performance (Killian et al., 2023; Shirali et al., 2023; Tang et al., 2022), but we focus on showing the limitations of offline RL under hidden confounding, and how our simple but effective penalty recovers good performance.

**Confounding in Medical Dataset.** As the aforementioned variables affect both the probability of treatment assignment and downstream patient evolution, they act as confounders over outcome models when excluded from the data. In Figure 6, we highlight how our delphic uncertainty measure captures confounded state-action pairs in concordance with the introduced confounders. Delphic uncertainty

---

[3]These therapeutic agents are commonly given to overcome shock in intensive care (Benham-Hermetz et al., 2012). Their administration strategy has already been studied as an RL task (Raghu et al., 2017).

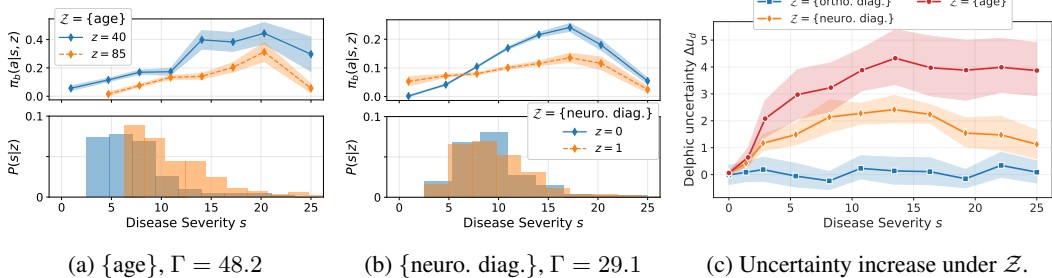

(a) {age}, $\Gamma = 48.2$     (b) {neuro. diag.}, $\Gamma = 29.1$     (c) Uncertainty increase under $\mathcal{Z}$.

Figure 6: **Delphic uncertainty as a function of** $(s, z)$ in real-world medical data, for $a = \{\text{vasopressors}\}$. In (a, b), we note the dependence of the behavioural policy $\pi_b$ (top) and/or state distribution $P(s)$ (bottom) on confounders $z$. In (c), delphic uncertainty increases most in confounded states and under factors with greater confounding strength, compared to orthopaedic diagnosis ($\Gamma = 3.4$).

| Confounders $\mathcal{Z}$ | $\Gamma$ | BCQ | BC | CQL | Delphic ORL |
|---|---|---|---|---|---|
| All 14 | $\approx 200$ | $54.6 \pm 1.3$ | $59.6 \pm 0.8$ | $59.3 \pm 0.9$ | $\mathbf{62.2 \pm 1.0}$ |
| {age} | 48.2 | $58.8 \pm 0.8$ | $64.7 \pm 0.5$ | $64.4 \pm 0.8$ | $\mathbf{66.5 \pm 0.9}$ |
| {neuro. diag.} | 29.1 | $55.0 \pm 1.3$ | $61.8 \pm 0.9$ | $59.6 \pm 1.7$ | $\mathbf{65.7 \pm 1.2}$ |
| {gastro. diag.} | 19.0 | $55.8 \pm 0.8$ | $60.9 \pm 0.6$ | $59.8 \pm 0.6$ | $\mathbf{63.3 \pm 1.1}$ |
| {trauma} | 16.3 | $56.3 \pm 0.8$ | $63.2 \pm 1.1$ | $63.5 \pm 0.7$ | $\mathbf{65.7 \pm 1.0}$ |
| {cardio. diag.} | 13.2 | $56.2 \pm 1.0$ | $60.6 \pm 0.7$ | $58.6 \pm 0.9$ | $\mathbf{62.7 \pm 1.1}$ |
| {hemato. diag.} | 11.6 | $59.6 \pm 0.9$ | $63.2 \pm 0.6$ | $63.1 \pm 0.7$ | $\mathbf{65.3 \pm 1.1}$ |
| {weight} | 8.3 | $60.1 \pm 0.8$ | $\mathbf{64.2 \pm 1.0}$ | $65.4 \pm 0.6$ | $\mathbf{66.3 \pm 0.9}$ |
| {sedation} | 6.8 | $61.2 \pm 0.8$ | $\mathbf{64.5 \pm 0.6}$ | $64.8 \pm 0.9$ | $\mathbf{65.3 \pm 1.2}$ |
| {endo. diag.} | 4.7 | $60.1 \pm 1.1$ | $63.1 \pm 0.6$ | $\mathbf{65.5 \pm 0.8}$ | $\mathbf{65.7 \pm 1.0}$ |
| {resp. diag.} | 4.4 | $61.6 \pm 1.3$ | $\mathbf{64.0 \pm 0.9}$ | $\mathbf{65.9 \pm 1.0}$ | $\mathbf{64.7 \pm 1.0}$ |
| {ortho. diag.} | 3.4 | $62.3 \pm 0.8$ | $\mathbf{64.6 \pm 0.6}$ | $\mathbf{65.8 \pm 0.7}$ | $\mathbf{65.9 \pm 1.0}$ |
| {surgical status} | 3.2 | $62.2 \pm 1.1$ | $64.3 \pm 0.5$ | $\mathbf{67.4 \pm 0.7}$ | $\mathbf{66.8 \pm 1.1}$ |
| {sepsis} | 2.8 | $60.3 \pm 0.9$ | $63.9 \pm 0.6$ | $\mathbf{65.4 \pm 0.7}$ | $\mathbf{66.2 \pm 1.0}$ |
| {intoxication} | 1.2 | $62.3 \pm 0.9$ | $63.4 \pm 0.5$ | $\mathbf{65.2 \pm 0.6}$ | $\mathbf{66.6 \pm 1.1}$ |
| $\emptyset$ | 1 | $62.6 \pm 0.8$ | $\mathbf{65.4 \pm 0.5}$ | $\mathbf{68.2 \pm 0.7}$ | $\mathbf{67.6 \pm 1.1}$ |

Table 2: **Off-Policy Evaluation (OPE)** on the real-world medical dataset. Delphic ORL yields improvements when $z$ strongly confounds treatment decisions (large $\Gamma$). Mean and 95% CIs over 10 runs. Best and overlapping results in bold.

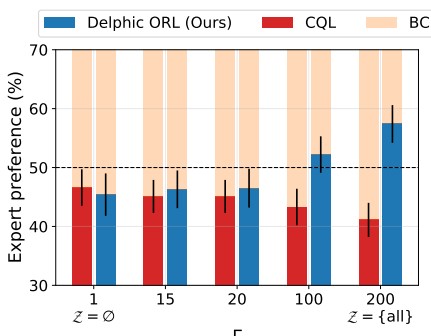

Figure 7: **Expert Clinician Evaluation** of treatment policies, supporting the conclusion that Delphic ORL improves learning in confounded settings.

is generally highest for high disease severity, where important factors such as age or comorbidity may affect the choice of treatment intensity (Azoulay et al., 2009). Indeed, delphic uncertainty increased to a greater extent under important confounders (e.g., age or patients' neurological diagnosis) than less critical factors (e.g., orthopaedic diagnosis).

**Confounding-Averse Policies.** We investigate the efficacy of our penalisation approach in learning policies that optimise patient outcomes in the presence of confounding bias. To evaluate the performance of these policies, we employ doubly-robust off-policy evaluation (OPE) (Jiang and Li, 2016; Le et al., 2019), which provides a confounding-independent estimate of treatment success by leveraging access to $z$. We refer the reader to Appendix D.2 for an exhaustive overview of this evaluation method. Table 2 shows our approach maintains improved performance even as the confounding level increases, while offline RL methods suffer from bias and yield suboptimal policies. As an ablation, we also studied the discrepancy of our trained policy with that in the data. We compared actions taken by our policy and to those in the data and found that our policy was more different than behaviour cloning, but more similar than offline RL baselines. This suggests our learned policy was indeed able to exploit the reward signal in the data, identifying observed treatment strategies that may be more robust to confounding biases. We refer the reader to Appendix E for an overview of this ablation.

**Expert Clinician Evaluation.** Motivated by the observed success of our method, we evaluated our algorithm using expert clinicians. Figure 7 shows the evaluation results of six human expert clinicians, who ranked pairs of different policies based on their observed patient outcomes. More specifically, the human experts were shown simulated patient trajectories and were asked to blindly compare the expected value of actions from either our policy or the CQL policy to those of the behaviour cloning policy. The results provide additional validation for the performance improvements of our method in confounded settings. We refer the reader to Appendix D.2 for an exhaustive overview of the clinician evaluation experiment.

# 7 RELATED WORK

Online-RL methods rely on environment interaction for training, limiting their applicability in many real-world domains such as healthcare (Gottesman et al., 2018). This has fueled research efforts in offline methods to optimise policies through pessimism (Buckman et al., 2021; Cheng et al., 2022; Jin et al., 2021; Levine et al., 2020; Uehara and Sun, 2022; Xie et al., 2021). Recent practical algorithmic developments in offline RL focus on addressing statistical errors induced by epistemic and aleatoric uncertainty (Fujimoto and Gu, 2021; Kidambi et al., 2020; Kostrikov et al., 2021; Kumar et al., 2020; Yu et al., 2020).

Structural errors such as confounding bias are also pervasive in offline RL (Lu et al., 2018a), but cannot be captured by epistemic or aleatoric uncertainty quantification methods, as they do not depend on data quantity. Confounding bias cannot be reduced to the missing information problem in partially-observable environments either (Hausknecht and Stone, 2015). History-dependent policies, for example, are equally prone to this source of error: while long-term information can recover latent environment information, it exacerbates distribution shifts between behavioural and learned policies when learning from observational data (Ortega et al., 2021; Swamy et al., 2022).

Several approaches have been proposed to address confounding bias in offline RL, summarised in Table 1. Most make assumptions to estimate the confounding variables, including access to the environment (Lu et al., 2018a; Zhang and Bareinboim, 2020) or to observable back- or front-door proxy variables (Kumor et al., 2021; Lu et al., 2022; Shi et al., 2022; Wang et al., 2021). This allows algorithms to apply covariate adjustment methods (Pearl, 2009) to correct for confounding when modelling alternative policies (interventional probabilities and counterfactuals). Variational approaches to modelling confounding distributions have been adopted in prior work addressing confounding bias (Lu et al., 2018a; 2022; Shi et al., 2022). These methods are sensitive to modelling assumptions in the non-identifiable settings (Rissanen and Marttinen, 2021), which further supports our motivation to not focus on estimating the true confounder distribution, but rather obtain a *plausible set* of confounder models compatible with the observational data.

Extensive work also discusses confounding bias and makes use of the aforementioned solutions in off-policy evaluation (Bennett and Kallus, 2021; Bennett et al., 2021) and bandits (Chen et al., 2023; Sen et al., 2017; Tennenholtz et al., 2021), but the proposed approaches remain poorly translatable to the practical optimisation challenge of learning offline RL policies.

Our work is also closely related to research on sensitivity analysis for treatment effect estimation under hidden confounding (Jesson et al., 2021; Kallus et al., 2019; Oprescu et al., 2023; Rosenbaum, 2002). These works propose partial identification bounds for confounded heterogeneous treatment effect estimation or bandit decision-making problems (Kallus and Zhou, 2018) by assuming a bound on the dependence of the behavioural policy on hidden confounders. In this context, Jesson et al. (2021) also distinguish sources of aleatoric and epistemic uncertainty from confounding biases. Other work has proposed sensitivity analysis bounds for off-policy evaluation, formulating uncertainty sets over policy returns (Kallus and Zhou, 2020; Namkoong et al., 2020; Zhang and Bareinboim, 2019). Still, regret bounds from sensitivity analysis remain wide and often ill-adapted to high-dimensional state and action spaces or sequential decision-making problems. Our approach thus complements these theoretical frameworks with a practical solution to addressing confounding bias in offline RL.

# 8 CONCLUSION

We proposed a practical solution to address the challenge of learning from confounded data, specifically in situations where confounders are unobserved and cannot be identified. Delphic ORL captures uncertainty by modelling world models compatible with the observational distribution, achieving improved performance across both simulated and real-world confounded offline RL tasks. Our results demonstrate that Delphic ORL can learn useful policies in cases where traditional algorithms fail due to excessive confounding. Overall, we believe research into tackling hidden confounding in offline RL will lead to more reliable and effective decision-making tools in various critical fields.

As limitations of our work, note that our evaluation focuses on medically-motivated scenarios, on the hypothesis that these are representative of other confounded decision-making contexts. Second, the modelling cost of compatible worlds $\mathcal{W}$ may be expensive for large-scale problems. We refer the reader to Appendix A.3 for further discussion on the limitations and possible impact of our work.

**Ethics Statement**    We propose a in-depth discussion on the limitations and possible societal impact of our work in Appendix A.3. We also provide more details on our approach to evaluating our algorithm using expert clinicians in Appendix D.2.

**Reproducibility Statement**    Implementation details for our approach and for baselines considered are presented in Appendix C. Dataset details and processing steps are included in Appendix D.1. We include source code as supplementary material.

**Acknowledgments**    We thank the anonymous reviewers for their thoughtful feedback. This project was supported by grant #2022-278 of the Strategic Focus Area "Personalized Health and Related Technologies (PHRT)" of the ETH Domain (Swiss Federal Institutes of Technology) and by ETH core funding (to Gunnar Rätsch). This publication was made possible by an ETH AI Center doctoral fellowship to Alizée Pace.

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

# A  ADDITIONAL RELATED WORK

## A.1  STATISTICAL UNCERTAINTY ESTIMATION

Uncertainty estimation is a crucial aspect of machine learning models, as it provides valuable insights into the reliability and confidence of model predictions and can be used to guide policy optimisation in reinforcement learning. Statistical sources of error can be estimated through aleatoric and epistemic uncertainty, which have been widely studied in the machine learning literature (Hüllermeier and Waegeman, 2021). In this section, we review existing methodologies for capturing and quantifying these two types of uncertainty.

**Aleatoric Uncertainty.**  Aleatoric uncertainty, also known as data uncertainty or irreducible noise, stems from the inherent variability and randomness in the observed data (Hüllermeier and Waegeman, 2021). This form of statistical uncertainty cannot be reduced even with infinite data quantities.

The most common approach to modelling aleatoric uncertainty is to set a probability distribution over model outputs and to learn its parameters (Kendall and Gal, 2017). Outputs can for instance be assumed to be normally distributed with either a fixed variance (introducing a single parameter to be estimated through maximum likelihood), or a variance that depends on the input. In this latter case of heteroscedastic aleatoric uncertainty, a separate neural network branch can be trained to predict the variance (Kendall and Gal, 2017).

**Epistemic Uncertainty.**  Epistemic uncertainty arises from the lack of knowledge or ambiguity in the model parameters (Hüllermeier and Waegeman, 2021), which can be reduced with additional data. Capturing epistemic uncertainty is particularly important to approximate model error in out-of-distribution scenarios (Jin et al., 2021).

Bayesian neural networks (BNNs) offer a principled approach to capturing epistemic uncertainty (Neal, 2012). By placing prior distributions over the model weights and using Bayesian inference, BNNs can provide posterior distributions over the weights, which represent the uncertainty in the model parameters. This uncertainty can then be propagated through the network to obtain predictive distributions that quantify epistemic uncertainty.

Bootstrap ensemble methods are another effective epistemic uncertainty estimation technique (Efron, 1982). These methods rely on creating multiple subsets, or bootstrapped samples, from the original dataset by randomly sampling with replacement. Each bootstrapped sample is then used to train a separate model, resulting in an ensemble of models with slightly different parameter configurations. By aggregating the predictions from these diverse models, the epistemic uncertainty can be estimated through measures such as variance or entropy. Bootstrap ensemble methods provide a practical and scalable approach to capturing model uncertainty, particularly when Bayesian methods are computationally expensive or infeasible (Lakshminarayanan et al., 2017).

Monte Carlo dropout sampling (Gal and Ghahramani, 2016) can also be used to estimate epistemic uncertainty by performing multiple forward passes with dropout enabled at test time. The distribution of predictions from these multiple samples gives an estimate of the predictive uncertainty. Finally, more recent efforts in epistemic uncertainty estimation include randomised priors (Osband et al., 2018), epistemic neural networks (Osband et al., 2022) and deep ensembles trained with Stein variational gradient descent (D'Angelo et al., 2021; Liu and Wang, 2016).

## A.2  NONIDENTIFIABLE CONFOUNDING BIAS

**Comparison to Sensitivity Analysis.**  In the causal inference literature, sensitivity analysis studies the robustness of treatment effect estimation to hidden confounding. This framework assumes a bound on the ratio between treatment propensities between any two confounder values (Rosenbaum, 2002) or on the ratio between treatment propensities when accounting for and marginalising confounders (Jesson et al., 2020; Kallus et al., 2019; Oprescu et al., 2023).

In contrast, the main assumptions in our delphic uncertainty estimate determine which 'world models' compatible with the observational data are considered to construct uncertainty sets over a given outcome model (Conditional Average Treatment Effect or, in the sequential setting, Q-value function). In particular, we consider a set of possible $\mathcal{Z}$ and prior distributions $p(z)$, and specify a model architecture for the dependence of the behavioural policy and transition, reward or value function on $z$ (which is then trained to fit the marginalised observational trajectory distribution).

Importantly, sensitivity analysis approaches require domain expertise to set maximum propensity-ratio parameter $\Gamma$ (Kallus et al., 2019), from which uncertainty sets over the modelled outcome are derived. Delphic ORL does come with its own set of hyperparameters (number of world models considered in $u_d$, pessimism hyperparameter $\lambda$), which can be determined through practical, quantitative means with arguably less domain expertise.

**Time-Varying Confounders.** The Contextual Markov Decision Process (Hallak et al., 2015) and associated problem described in Section 2 describes confounders as sampled from a context distribution $\nu(z)$ and fixed over the course of an episode. We note that this framework does not exclude the existence of time-varying confounders. Consider the Markov Decision Process with Unmeasured Confounding (MDPUC) (Zhang and Bareinboim, 2016), in which a new i.i.d. hidden confounder variable $z_t$ affects the transition at each timestep $t$. This framework can be framed as a CMDP where the overall episode context $z = \{z_1, \ldots, z_H\}$ includes all confounder variables. Although we do not focus on this specific framework in our experimental setting, this would form an interesting avenue for further work. An important distinction between MDPUC and Partially-Observable Markov Decision Processes (POMDPs) is the assumption that confounder variables are sampled i.i.d. at each timestep. While POMDPs can therefore be viewed as the generalisation of this decision-making setup, note that confounding biases are only induced in this setup if the behavioural policy has access to some missing information about the state variable.

Finally, note that Saengkyongam et al. (2023); Tennenholtz et al. (2022) also study confounding in offline environments, but are more concerned with the complementary challenge of covariate shift – with the latter work even assuming access to the contextual information.

### A.3 BROADER IMPACT, LIMITATIONS AND FUTURE WORK

Addressing hidden confounding in offline reinforcement learning has the potential to significantly impact the development and deployment of decision support systems in real-world applications. In real-world data, the behaviour policy is typically that of a human's, who make use of a rich density of information to take decisions, with important factors likely not all recorded into our dataset. By improving the validity of causal conclusions drawn from data, Delphic ORL can improve the effectiveness and safety of RL-based decision-making in critical fields. For example, learning to drive autonomously from observational data could benefit from delphic uncertainty (Haan et al., 2019), as human drivers rely on many unobserved sources of information, such as road conditions, that affect the safety of their behaviour. In digital marketing, another common use of reinforcement learning in industry (Thomas et al., 2017), learning from observational campaigns without A/B testing can also be prone to confounding. Finally, emerging decision support systems in justice (Gans-Combe, 2022), hiring (Sánchez-Monedero et al., 2020) or education (Singla et al., 2021) are likely to suffer fromm hidden confounding biases if unaccounted for.

While our results demonstrate the efficacy of Delphic ORL in learning useful policies in the presence of confounding, it is important to acknowledge the limitations and potential unintended consequences associated with RL algorithms, especially in high-stakes applications such as healthcare. Collaboration with domain experts is crucial to ensure thorough evaluation of RL algorithms (Gottesman et al., 2018). In clinical settings, predictive or recommendation models derived from Delphic ORL should not be solely relied upon, and mitigation strategies must be implemented to minimise negative consequences during deployment.

An important consideration in the application of Delphic ORL is the trade-off between confounding bias and estimation variance in Q-function estimation, as noted in other work addressing confounding bias (Wang and Blei, 2019). This emphasizes the significance of large, high-quality training datasets to leverage the benefits of Delphic ORL and ensure sufficient predictive power.

As our experiments primarily focused on medically-motivated confounding scenarios, future work should investigate the applicability and generalisation of Delphic ORL to other domains. Although our framework does not in theory exclude dynamic environments where confounding factors change over time (see Appendix A.2), an empirical study of the behaviour of delphic uncertainty estimates and pessimism penalties may reveal new challenges in this context.

Finally, the question of how to best approximate the set of compatible worlds $\mathcal{W}$ in Definition 4.1 remains open. In Section 5 and Appendix C, we detail our approach which efficiently captures variability across counterfactual value-function, but further theoretical or practical work on how

best to model $\mathcal{W}$ would likely improve the calibration of delphic uncertainty estimates. Better approximation algorithms may also improve the efficiency, scalability, and modelling power of our method for very high-dimensional, highly confounded problems – although our real-world data analysis forms a promising first proof-of-concept.

# B  THEORETICAL DETAILS

## B.1  PROOF OF THEOREM 4.2

We start by considering the decomposition of variance in $Q_\theta^\pi$ caused by random variable $\theta$. In the following, we drop superscript $\pi$ for clarity.

First, we decompose $\mathrm{Var}(Q_\theta \mid \theta)$:

$$
\begin{aligned}
\mathrm{Var}(Q_\theta \mid \theta) &= \mathbb{E}[Q_\theta^2|\theta] - \mathbb{E}[Q_\theta|\theta]^2 \\
\mathbb{E}_\theta[\mathrm{Var}(Q_\theta \mid \theta)] &= \mathbb{E}_\theta[\mathbb{E}[Q_\theta^2|\theta]] - \mathbb{E}_\theta[\mathbb{E}[Q_\theta|\theta]^2] \\
&= \mathbb{E}[Q_\theta^2] - \mathbb{E}_\theta[\mathbb{E}[Q_\theta|\theta]^2]
\end{aligned}
\tag{3}
$$

where the last line results from the law of iterated expectations: $\mathbb{E}_B[\mathbb{E}[A|B]] = \mathbb{E}[A]$ for two random variables $A, B$.

Next, we study $\mathrm{Var}(\mathbb{E}[Q_\theta \mid \theta])$:

$$
\begin{aligned}
\mathrm{Var}_\theta(\mathbb{E}[Q_\theta \mid \theta]) &= \mathbb{E}_\theta[\mathbb{E}[Q_\theta \mid \theta]^2] - \mathbb{E}_\theta[\mathbb{E}[Q_\theta \mid \theta]]^2 \\
&= \mathbb{E}_\theta[\mathbb{E}[Q_\theta \mid \theta]^2] - \mathbb{E}[Q_\theta]^2
\end{aligned}
\tag{4}
$$

again using iterated expectations.

Summing equations 3 and 4, we obtain:

$$
\begin{aligned}
\mathbb{E}_\theta[\mathrm{Var}(Q_\theta \mid \theta)] + \mathrm{Var}_\theta(\mathbb{E}[Q_\theta \mid \theta]) &= \mathbb{E}[Q_\theta^2] - \mathbb{E}[Q_\theta]^2 \\
&= \mathrm{Var}(Q_\theta)
\end{aligned}
\tag{5}
$$

This result is known as the law of total variance (Weiss et al., 2006), which can be interpreted as a decomposition of epistemic and aleatoric uncertainty (Kendall and Gal, 2017).

We can rewrite the above result within a given world model $w$, denoting $\theta$ as $\theta_w$. Now conditioning on the world model $w$, we have:

$$
\mathrm{Var}(Q_{\theta_w} \mid w) = \mathbb{E}_{\theta_w}[\mathrm{Var}(Q_{\theta_w} \mid \theta_w, w)|w] + \mathrm{Var}_{\theta_w}(\mathbb{E}[Q_{\theta_w} \mid \theta_w, w]|w)
\tag{6}
$$

We also write equation 5 such that the conditioning random variable is now $w$, which induces variation in $Q_{\theta_w}$ if we consider a counterfactual trajectory distribution. Combined with Equation (6), we obtain:

$$
\begin{aligned}
\mathrm{Var}(Q_{\theta_w}) &= \mathbb{E}_w[\mathrm{Var}(Q_{\theta_w}|w)] + \mathrm{Var}_w\left(\mathbb{E}[Q_{\theta_w} \mid w]\right) \\
&= \mathbb{E}_w\left[\mathbb{E}_{\theta_w}[\mathrm{Var}(Q_{\theta_w} \mid \theta_w, w)|w] + \mathrm{Var}_{\theta_w}(\mathbb{E}[Q_{\theta_w} \mid \theta_w, w]|w)\right] \\
&\qquad\qquad + \mathrm{Var}_w\left(\mathbb{E}_{\theta_w}[\mathbb{E}[Q_{\theta_w} \mid \theta_w, w]|w]\right)
\end{aligned}
\tag{7}
$$

using iterated expectations. This concludes the proof of Theorem 4.2.

Note that if we assume $Q_{\theta_w}$ has a Gaussian distribution for fixed $\{\theta_w, w\}$, parameterised as $\mathcal{N}(\mu_{\theta_w}, \sigma_{\theta_w}^2)$, we have $\mathrm{Var}(Q_{\theta_w} \mid \theta_w, w) = \sigma_{\theta_w}^2$ and $\mathbb{E}[Q_{\theta_w} \mid \theta_w, w] = \mu_{\theta_w}$. We obtain results in Equation (2) by expanding the first term in the variance decomposition, $\mathbb{E}_{\theta_w}[\sigma_{\theta_w}^2|w]$, as follows:

$$
\begin{aligned}
\mathbb{E}_{\theta_w}[\sigma_{\theta_w}^2|w] &= \mathbb{E}_{\theta_w}[\sigma_{\theta_w}^2|w] - \mathbb{E}_{\theta_w}[\sigma_{\theta_w}|w]^2 + \mathbb{E}_{\theta_w}[\sigma_{\theta_w}|w]^2 \\
&= \mathrm{Var}_{\theta_w}(\sigma_{\theta_w}|w) + \mathbb{E}_{\theta_w}[\sigma_{\theta_w}|w]^2.
\end{aligned}
$$

### B.2 ASYMPTOTIC INTERPRETATION OF THEOREM 4.2

We consider three extreme cases of Theorem 4.2 to clarify its decomposition. First, we consider the limit of infinite-data with no confounding (e.g., no dependence on $z$). In this case, $\theta_w$ and $w$ converge to a single ground-truth. Any remaining statistical error will come from the intrinsic environment stochasticity or the behavioural policy, and therefore has an aleatoric nature. Indeed, only the first term in Theorem 4.2 would remain.

Next, consider the setting in which the value is a deterministic mapping of states, with only one compatible world model. Learning from finite data quantities leads to statistical error in optimising the parameters $\theta_w$, and is known as epistemic uncertainty. Indeed, deterministic environments with only one compatible world model will reduce Theorem 4.2 to the second term.

Finally, we consider the case of infinite data in a deterministic setting. In this case, multiple compatible world models may exist which induce the same observational distribution (as demonstrated in Section 3). The source of error remaining is delphic uncertainty, and arises if multiple models assign high likelihood to the observational data, but return different estimates of the value. In this paper we propose to estimate this final form of uncertainty by learning an ensemble of compatible world models, in a similar fashion to the bootstrap method for quantifying epistemic uncertainty.

## C IMPLEMENTATION DETAILS

### C.1 STATISTICAL & DELPHIC SOURCES OF UNCERTAINTY

**World Model Training.** We implement world models as variational models for estimating the confounder distribution, jointly with a model for the behaviour policy $\pi_b$ and for the action-value function $Q^{\pi_b}$, both dependent on a $z$ sampled from the posterior. As the environments we consider have discrete action spaces, we learn the behaviour policy by minimising its cross-entropy on the training data, as in behaviour cloning. Training is carried out for 50 epochs or until loss on the validation subset (10% of training data) increases for more than 5 consecutive epochs. Within a world model $w$, hyperparameters $\{\alpha, \beta\}$ can be tuned based on prediction performance on the validation set.

Model $Q^{\pi_b}$ corresponds to an on-policy action-value function approximation. We compute targets through Monte Carlo updates (for the sepsis environment with sparse episodic rewards) or Temporal Difference learning (for the real-world ICU dataset) based on samples from the observational training data with a discount factor of $\gamma = 0.99$. The Q-function is trained as a classifier over 200 quantiles.

Between world models $w$, the confounder space dimensionality is randomly varied over $|\mathcal{Z}| = \{1, 2, 4, 8, 16\}$, and the prior for $p(z) = \mathcal{N}(z; 0, \Sigma^2)$ is randomly varied through the variance for each $z$-dimension, $\Sigma_{ii}^2 = \{1.0, 0.1, 0.01\}$. For the sepsis simulation, the encoder architecture for the confounder distribution $\nu(z|\tau)$ consists of a multi-layer perceptron with hidden layer dimensions $(128, 64, 32)$ and ReLU activation before the final layer mapping to dimension $|\mathcal{Z}|$. For the real dataset, the encoder architecture is implemented as a transformer (Vaswani et al., 2017) with 2 layers, 4 heads, and embedding dimension 32, considering a maximum history length of 10 tokens. The behavioural policy $\pi_b(a|s, z)$ and action-value function $Q^{\pi_b}(s, a, z)$ are both implemented as multilayer perceptrons with hidden layer dimensions $(32, 64, 128)$ and ReLU activation.

**Uncertainty Estimates.** Additional inductive biases can be incorporated to capture epistemic and aleatoric uncertainty within a single world model $w$, as these relate to statistical sources of uncertainty. Following prior work (Kendall and Gal, 2017; Yu et al., 2020), we capture aleatoric uncertainty by modelling a normal probability distribution over outputs $(\pi_b, Q^{\pi_b})$. We then measure epistemic uncertainty within each world model $w$ by training on different data bootstraps, returning an ensemble of parameters $\{\theta_w^1, \theta_w^2, \ldots\}$ for each $w$.

Recalling Equation (2), the delphic uncertainty term $\mathrm{Var}_w(\mathbb{E}_\theta[\mu_{\theta_w}])$ is estimated by **measuring the variance between predictions $\mu_{\theta_w}$ (averaged over model parameters $\theta_w$), across across multiple generative models** $w$. Epistemic uncertainty can be estimated as the variance of outputs over different model parameters $\theta_w$, averaged across worlds $w \in \mathcal{Z}$. Finally, aleatoric uncertainty is measured through the fitted probability distribution over model outputs $Q_{\theta_w}^{\pi_b}$, averaged over all $\theta_w$ in a given world, and over all worlds $w \in \mathcal{W}$.

---

**Algorithm 2:** Delphic Offline Reinforcement Learning: Bellman Penalty in Offline Q-Learning Algorithm.

---

1 **Input:** *Observational dataset $\mathcal{D}$, Model-free Offline RL algorithm (e.g. CQL (Kumar et al., 2020)), Penalty hyperparameter $\lambda$.*

2 Learn a set of compatible world models $\{\mathcal{Z}_w, \nu_w, \rho_{0,w}, P_{r,w}, T_w, \pi_{b,w}\}_{w \in \mathcal{W}}$ that all factorise to $P_{\pi_b}(\tau)$.

3 Obtain counterfactual predictions $Q_w^\pi$ for each $w \in \mathcal{W}$.

4 Define local delphic uncertainty: $u_d^\pi(s, a) = \mathrm{Var}_w(Q_w^\pi(s, a))$.

5 Initialise Q-function parameters $\phi$.

6 **for** *each iteration* **do**

7     Sample $(s, a, r, s') \sim \mathcal{D}$.

8     Compute penalised Bellman target: $Q'_{target} = r + \gamma \max_{a' \in \mathcal{A}} Q_\phi(s', a') - \lambda u_d^\pi(s, a)$, where $\pi(a|s) = \mathrm{argmax}_a Q_\phi(s, a)$.

9     Perform gradient descent w.r.t. $\phi$ on $\left[Q_\phi(s, a) - Q'_{target}(s, a)\right]^2 + \mathcal{R}_{offline}(\phi)$, where regularisation term $\mathcal{R}_{offline}$ depends on the choice of offline learning algorithm.

10 **end**

---

The number of world models $W$ was varied between 5 and 20 for both datasets and was chosen as the smallest number converging to an average delphic uncertainty comparable to the largest $W$. An ablation of delphic uncertainty as a function of the number of world models is given in Appendix E. This resulted in 10 and 15 world models for the sepsis and real-world datasets respectively. Finally, each world model was trained over 5 different data bootstraps to estimate epistemic uncertainty. Overall, compared to sensitivity analysis where parameter $\Gamma$ needs to be fixed through domain expertise (Oprescu et al., 2023), we found delphic uncertainty to be less dependent on expert input in determining model parameters.

**Counterfactual Estimates.** Our approach is to change the policy term in $P_{\pi_b}$ to obtain counterfactual estimates, using the importance sampling estimator $Q_{\theta_w}^\pi(s, a) = \mathbb{E}_{\tau \sim \mathcal{D}} \mathbb{E}_{z \sim \nu_{\theta_w}(z|\tau)} \left[ \frac{\pi(a|s)}{\pi_{b,\theta_w}(a|s,z)} Q_{\theta_w}^{\pi_b}(s, a, z) \right]$. Note that other factors in the world model (e.g. $\nu_w$, $Q_w^{\pi_b}$) could be varied to obtain general counterfactual predictions in this world model. As an example, we also found promising results by measuring delphic uncertainty through variation across $w$ over the following counterfactual quantity: $\mathbb{E}_{(s,a) \in \mathcal{D}} \mathbb{E}_{z \sim p_w(z)} \mathbb{E}_{\theta_w} [Q_{\theta_w}^{\pi_b}(s, a, z)]$, where $z$ is sampled from the model prior $p_w(z)$ instead of the learned posterior $\nu_{\theta_w}(z|\tau)$. In this case, the resulting delphic uncertainty estimate, capturing variation over the counterfactual quantity across world models, becomes independent of a given policy – and dependent on the new quantity introduced (in the previous example, on prior $p_w(z)$).

### C.2 DELPHIC OFFLINE REINFORCEMENT LEARNING

We detail our learning procedure in Algorithm 2. As our base offline RL algorithm is CQL (Kumar et al., 2020), our regularisation term $\mathcal{R}_{offline}(\phi)$ is the CQL penalty: $\mathcal{R}_{offline}(\phi) = \alpha \left[ \log \sum_{\tilde{a} \in \mathcal{A}} \exp Q_\phi(s, \tilde{a}) - Q_\phi(s, a) \right]$. We base our algorithm on an existing implementation for CQL (Seno and Imai, 2022), which includes additional training details for stability, such as target networks, double Q-networks and delayed updates (Fujimoto et al., 2018). For architecture details, see the baseline implementation of CQL in Appendix C.3. As for all baseline algorithms, we train for 100 epochs, using 500 (sepsis dataset) or $10^4$ (ICU dataset) timesteps per epoch. In practice, the policy $\pi$ considered for uncertainty estimation and the target network are updated every 8000 timesteps, to improve stability in training.

Note that an actor-critic variant of Algorithm 2 is also feasible, setting $\pi$ in $u_d^\pi$ to be the actor policy, as well as other offline learning paradigms in $\mathcal{R}_{offline}(\phi)$, such as BC regularisation (Fujimoto and Gu, 2021).

**Alternative Forms of Pessimism.** Following the discussion on alternative forms of pessimism in Section 5.2, we propose practical alternatives to the Delphic ORL penalty in Line 8 of Algorithm 2, which substracts a factor of $u_d$ from the Q-function Bellman target based. In the following, note that $u_d$ can also be independent of $\pi$ if varying over different factors in $P_{\pi_b}$ as detailed above.

- **Delphic ORL via Uncertainty Threshold:** One approach, inspired by Batch Constrained Q-Learning (Fujimoto et al., 2019), is to constrain value function updates to only consider actions falling below a certainty uncertainty threshold. For a tuple $(s, a, r, s')$, the Q-function Bellman target can be computed as: $Q'_{target} = r + \gamma \max_{a' : u_d^\pi(s', a') < \lambda} Q_\phi(s', a')$, where $\lambda$ is a threshold controlling the maximum delphic uncertainty accepted for a given action choice.

- **Model-Based Delphic ORL:** In model-based methods, a penalty proportional to the uncertainty $u_d(s, a)$ can be substracted from the reward function $r(s, a)$, as in Yu et al. (2020). The effective reward function becomes: $\tilde{r}(s, a) = r(s, a) - \lambda u_d(s, a)$.

- **Delphic ORL via Weighting:** The uncertainty measure can also be used to weight samples in the objective function, prioritising unconfounded states and actions during training:

$$\mathbb{E}_{(s,a,r) \sim \mathcal{D}} \left[ \frac{\lambda}{u_d(s, a)} \mathcal{L}(s, a, r) \right]$$

where $\mathcal{L}$ can be the Q-function Bellman update or the supervised learning objective for behaviour cloning.

We compare the performance of different implementations of pessimism on the simulated sepsis environment in Appendix E. Finally, as delphic ORL benefits from full trajectories in world model training, we also measure whether introducing history-dependence can help our offline RL framework. We achieve this by modifying our implementation of CQL to incorporate a representation of history in the Q-function and/or the policy networks, extracted through a transformer.

**Hyperparameter Tuning.** There is no natural validation criterion in Offline RL, and the best approach to choose hyperparameters in this context remains an open question (Levine et al., 2020). In practice, we run our algorithm for 4 different values of $\lambda \in \{10^{-3}, 10^{-2}, 10^{-1}, 1\}$ and choose the final policy giving the best off-policy evaluation performance on the validation set (using the Fitted Q-Evaluation implementation available in the codebase, Le et al. (2019)). As noted in related works, expert input may be useful at this stage to also determine how strong a penalty again potential hidden confounding would be desirable or how much confounding could be expected (Rosenbaum, 2002). Other hyperparameters specific to offline RL algorithms are tuned in the same way and are given in the following section.

### C.3 Baseline Methods & Training Details

All reinforcement learning algorithms and baselines are implemented based on the open access `d3rlpy` library (Seno and Imai, 2022). The discount factor used is $\gamma = 0.99$, and state and actions are normalised to mean 0 and variance 1 (Fujimoto and Gu, 2021) for all algorithms. Training is carried out on NVIDIA RTX2080Ti GPUs on our local cluster, using the Adam optimiser with default learning rate and a batch size of 32. Models are trained for 100 epochs with 500 (sepsis dataset) or $10^4$ (ICU dataset) timesteps per epoch. Model-specific hyperparameters are tuned as in Delphic ORL.

**Behaviour Cloning (BC).** Behaviour cloning (Ross et al., 2011) is a supervised learning model of the behaviour policy, mapping states to actions observed in the dataset. After considering the following architectures: multi-layer perceptron (MLP), Long Short Term Memory (LSTM) network (Hochreiter and Schmidhuber, 1997), Gated Recurrent Unit (GRU) (Cho et al., 2014) and Transformer (Vaswani et al., 2017), GRU was found to give the best validation performance on both the simulated and real datasets. Implementation details for the GRU BC models include two hidden layers of dimension $(64, 32)$ and ReLU activation. The last layer is passed through a softmax layer to produce action probability outputs, and the model is trained by minimising action cross-entropy over the observational dataset, with L2 regularisation of weight 0.01.

**Conservative Q-Learning (CQL).** Discrete CQL (Kumar et al., 2019) is implemented with a penalty hyperparameter $\alpha$ of 1.0 (sepsis environment) and 0.5 (ICU dataset), tuned over the following values: $\{0.1, 0.5, 1.0, 2.0, 5.0\}$. The Q-function is implemented as a distributional model with a standard MLP architecture (two linear layers with 256 hidden units) and 200 quantile regression outputs.

**Batch Constrained Q-Learning (BCQ).** Discrete BCQ (Fujimoto et al., 2019) is implemented with a threshold for action flexibility set to 0.5 for both environments, tuned over the following values: $\{0.1, 0.3, 0.5, 1.0, 2.0, 5.0\}$. The Q-function is implemented as a distributional model with a standard MLP architecture (two linear layers with 256 hidden units) and 200 quantile regression outputs.

## D    EXPERIMENTAL DETAILS

### D.1    DECISION-MAKING ENVIRONMENTS

**Sepsis Environment.** Introduced by Oberst and Sontag (2019), this environment simulates the trajectory of patients in the intensive care. Based on the authors' publicly available code[4], our state space $\mathcal{S}$ consists of 4-dimensional observation vectors (measures for heart rate, systolic blood pressure, oxgenation and blood glucose levels) which we normalise to mean and variance $(0, 1)$. The discrete action space $\mathcal{A}$ comprises the combination of three binary treatments (antibiotic, vasopressor or ventilation administration) for a total dimension of 8. An unobserved binary variable $z$ encodes the diabetic status of patients, with 20% of trajectories having a positive status. The agent obtains a reward of $+1$ if the patient reaches a healthy state (and is thus ready for discharge) and a negative reward of $-1$ if the patient reaches a death state.

The observational dataset $\mathcal{D}$ is generated by rolling out the optimal (diabetes-aware) policy in the environment for 10,000 environment interaction steps, taking a random action with probability $\epsilon = 0.1$ to ensure sufficient state-action coverage for offline learning. The maximum episode length is set to 20 timesteps. The resulting dataset has a confounding strength of $\Gamma = 100$.

Environment stochasticity can be varied by changing the variance around the originally deterministic reward obtained at the end of a trajectory, between $\sigma_r^2 = 0$ as in the original environment and $\sigma_r^2 = 0.4$. Datasets of varying confounding strength $\Gamma \in [1, 100]$ are obtained by setting the behaviour policy for $z = 1$ as a weighted average of the policies for different $z$ values: $(1 - p)\pi_b(z = 0) + p\pi_b(z = 1)$, where $p$ depends on $\Gamma$ and $\epsilon$. Environment transition and reward functions and their dependence on $z$ are kept fixed. Finally, we vary the dimension of the confounder space $\mathcal{Z}$ by introducing more binary indicators with the same effect on the transition dynamics as the diabetes indicator.

**Electronic Health Records Dataset.** Our real-world data experiment is based on the publicly available HiRID dataset (Hyland et al., 2020). This dataset counts over 33 thousand patient admissions at an intensive care unit in Bern University Hospital, Switzerland (Hyland et al., 2020) and can be pre-processed using open access code from the HiRID benchmark (Yèche et al., 2021). Patient stays were downsampled to hourly measurements and truncated to a maximum length of 20 hours and default training, validation and test sets were used.

We consider the task of optimising fluid and vasopressor administration ($\mathcal{A}$ is the combination of two binary choices). The reward function is designed to penalise circulatory failure events ($r = -1$ for all timepoints in the duration of the event) and to reward timepoints where the patient is not in such a critical state ($r = 1$, and $r = 2$ in the timepoint following recovery from circulatory failure). Circulatory failure events for each patient are labelled following internationally accepted criteria (Yèche et al., 2021). This short-term reward function is dense, unlike previous RL work on optimising intravenous fluid and vasopressor administration (Raghu et al., 2017), making off-policy evaluation more reliable (Gottesman et al., 2018).

The state space $\mathcal{S}$ consists of all variables in the electronic health records which are not considered treatment for the organ system considered, based on the variable categorisation released with the dataset (Hyland et al., 2020). This results in a state space dimensionality of 203. The list of variables excluded for each task in given in Table 3. At each timepoint within a patient stay, we also compute the Sequential Organ Failure Assessment (SOFA) score (Vincent et al., 1996) which is used to quantify the severity of a patient's illness in the intensive care unit. A higher score indicates greater severity of illness.

Selected confounders are obtained by excluding some state dimensions from the observational dataset (up to $|\mathcal{Z}| = 14$). These variables do not constitute the *entire* confounder space, as much exogenous, unrecorded information affects patient evolution and is taken into account in medical treatment

---

[4]https://github.com/clinicalml/gumbel-max-scm

Table 3: **Offline reinforcement learning task on real-world medical dataset.**

| Task | Circulatory treatment | |
|---|---|---|
| Action space $\mathcal{A} = \{0,1\}^2$ | Fluids | Vasopressors |
| Organ failure avoided by $R$ | Circulatory failure | |
| State space $\mathcal{S}$ (selected variables, $|\mathcal{S}| = 204$) | Heart rate
Body temperature
Blood pressure
Cardiac output
Oxygen saturation
Lactate
PaO2
Serum sodium
Haemoglobin
Other lab values
Antibiotics
Diuretics
Cerebrospinal fluid drain | Respiratory rate
Urinary output
GCS score
Central venous pressure
Base excess
Arterial pH
Creatinine
Serum potassium
Glucose
Ventilator settings
Steroids
Insulin
Anticoagulants
. . . |
| Other treatment variables excluded | Blood product infusions
Cristalloid infusion
Colloid infusion | Vasodilators
Antiarrhymic agents
Antihypertensive agents |
| Confounder variables $z$ | Age
Weight
Gastrointestinal diagnosis
Neurological diagnosis
Hematology diagnosis
Sedation
Emergency status | Cardiovascular diagnosis
Pulmonary diagnosis
Orthopaedic diagnosis
Metabolic/endocrine diagnosis
Trauma diagnosis
Intoxication
Surgical status |

decisions (Yang and Lok, 2018). We ignore this in our analysis as we cannot evaluate with respect to this missing information, but we note that this is precisely the motivation behind our work.

The confounding strength $\Gamma$ for each confounding space $\mathcal{Z}$ considered was estimated as follows. Each point in the training dataset was binned into a $(s, a, z)$ category, depending on its discrete action and context values $(a, z)$ and on its SOFA score as a summary variable for $s$. We discretise the SOFA score into 5 quantiles. Finally, we compute the mean policy value for each $(s, a, z)$ bin through $\pi_b(a|s, z) = P(a, s, z)/P(s, z)$, and we take $\Gamma$ as the ratio $\max_{z, z', s, a}[\pi_b(a|s, z)/\pi_b(a|s, z')]$.

## D.2 ANALYSIS DETAILS

In this section, we provide additional details pertaining to the analysis of our experimental results. All results reported in this work include 95% confidence intervals around the mean, computed over ten training runs unless otherwise stated. Environment returns and off-policy evaluation results are normalised on a scale of 0 to 100. Figure 4 was obtained by varying the dimension on the x-axis, while keeping the other variables fixed to $N = 864$ trajectories, confounding strength $\Gamma = 15$ and reward function variance $\sigma_r^2 = 0.0$. To generate Figure 6, patients with the relevant confounder ($z$) value were binned by disease severity and the probability of vasopressor prescription (top) and the overall density (bottom) in each group were computed. Figure 6c was then obtained by computing the relative increase in delphic uncertainty when including the relevant $z$-dimension to the hidden context space $\mathcal{Z}$ (in other words, removing this dimension from the visible state space).

**Off-Policy Evaluation (OPE).** Doubly robust methods trade off bias of an approximate reward model and of weighted methods with the high variance of importance sampling approaches (Jiang and Li, 2016). Assuming $z$ is accessible for each trajectory at *evaluation* time to overcome confounding, doubly-robust off-policy evaluation estimates the value of policy $\tilde{\pi}$ as follows:

$$V_{DR}(\tilde{\pi}) = \mathbb{E}_{(s,a,r,z) \in \mathcal{D}} \left[ \frac{\tilde{\pi}(a|s)}{\hat{\pi}_b(a|s, z)} \{r - Q(s, a, z)\} + Q(s, \tilde{\pi}(s), z) \right], \tag{8}$$

where $\hat{\pi}_b$ is a model for the behavioural policy and $Q$ for expected returns under $\tilde{\pi}$, learned on the dataset with observable $z$.

Fitted Q-Evaluation is an established value estimation method (Le et al., 2019). The algorithm iteratively applies the Bellman equation to compute bootstrapping targets for Q-function updates: $Q_{k+1} \leftarrow \arg\min_Q \mathbb{E}_{(s,a,r,z) \in \mathcal{D}} \left[ \{r - Q(s, a, z) + \gamma Q_k(s', \tilde{\pi}(s'), z)\}^2 \right]$ which can be solved as a supervised learning problem. This results in a learned Q-value for the evaluated policy $Q^{\tilde{\pi}}(s, a, z)$ which can be used in the weighted doubly-robust estimate in Equation (8) to provide return estimates in Table 2.

Both the Q-function and the behaviour policy in Equation (8) are parametrised as a fully-connected neural network dimension with 3 layers of hidden dimension (64, 32, 16) and ReLU activation. The former is trained by minimising the mean squared error with the Q-function update above, the latter by minimising the cross-entropy with respect to action choices in $\mathcal{D}$.

**Human Policy Evaluation.** Off-policy evaluation has limitations, being itself prone to its own set of statistical errors and data-related concerns (Gottesman et al., 2018). We aim to confirm conclusions drawn over OPE returns through a human expert evaluation of treatment policies.

Synthetic patient trajectories are first generated by randomly sampling from the ICU dataset along each state dimension, with varying amounts of contextual information as detailed in Table 4. Action choices at the end of the trajectories are computed for the Delphic ORL, CQL and BC policies, trained on the observational dataset with the same degree of confounding. Trajectories are selected if they induced a disagreement between these methods, to shed light on potential improvements or harmful behaviour learned by the offline RL models. Trajectories are then simplified into 12 critical variables (as shown in Figure 9), and shown to physicians, who are asked to rank two treatment options in terms of expected patient outcomes. Unknown to the physicians, and in a random order, one of the options was predicted by the Delphic ORL or CQL policy, and the other by the BC baseline. Overall, we consulted six clinicians with different degrees of expertise in intensive care (from junior assistant doctors to department heads) from Switzerland and the United Kingdom, collecting their treatment preferences over 45 such trajectories.

Table 4: **Data settings considered during expert clinician evaluation.** Physicians are asked to rank action choices based on only state information ($\Gamma \approx 200$), or with varying amounts of observed contextual information ($\{$All 14$\}$ refers to all possible $\mathcal{Z}$ variables outlined in Table 3).

| $\Gamma$ | **1** | **15** | **20** | **100** | **200** |
|---|---|---|---|---|---|
| $\|\mathcal{Z}\|$ | 0 | 10 | 11 | 13 | 14 |
| Observed | $\{$All 14$\}$ | $\{$Age, Neuro. diag., Trauma diag., Surgery$\}$ | $\{$Age, Neuro. diag. Surgery$\}$ | $\{$Age$\}$ | $\emptyset$ |

We contacted our local institution's ethics committee to enquire about the possible necessity of ethics approval for this experimental framework. We were informed that this was not considered necessary as the experts contribute to the validation of algorithms and are thus not themselves the subject of the research, and as the undertaking comes with minimal risks to those experts (anonymous data collection). Best practice was nonetheless observed, by providing participants with an information and consent letter (Figure 8) to inform them of their rights and obligations, and of how their data is collected and used. Participants were asked to read and sign this letter before collecting their anonymous expert opinion.

Results in Figure 7 report the preference of clinicians for actions from either Delphic ORL or CQL or from behaviour cloning. We note their overall preference for the Delphic ORL policy in the confounded settings (high $\Gamma$). As more contextual information about the patient becomes available, however, and confounding is less marked (small $\Gamma$), physicians favour the behaviour cloning policy – closer to expected clinical practice.

# E  ABLATIONS AND ADDITIONAL RESULTS

## E.1  CONTEXTUAL BANDIT ENVIRONMENT

We also conduct an experiment based on the illustrative example provided in Section 3. We define the confounded simulation environment as in World 2 (Figure 2c) and generate 5k pairs as observational data. We present our results in Table 5.

Table 5: **Comparison of offline and delphic offline RL** on contextual bandit experiment (Figure 2).

| Reward function | $a_0$ | $a_1$ | $a_2$ |
|---|---|---|---|
| $P_{\pi_b}(r\|a)$: Ground truth **observational** reward | **0.6** | 0.5 | **0.6** |
| $P_{\pi_b}(r\|a)$: Estimation $\pm$ epistemic uncertainty | $0.59 \pm 0.02$ | $0.47 \pm 0.02$ | $0.59 \pm 0.02$ |
| $\mathbb{E}_z[P(r\|a,z)]$: Ground truth **interventional** reward | 0.41 | **0.5** | 0.41 |
| $\mathbb{E}_z[P(r\|a,z)]$: Estimation $\pm$ delphic uncertainty | $0.46 \pm 0.29$ | $0.53 \pm 0.02$ | $0.59 \pm 0.28$ |

- The first two rows capture the ground truth and estimated reward function, based on the observational data. The latter is what would be optimised in RL or offline RL (with a penalty on epistemic uncertainty), resulting in a policy which chooses $a_0$ or $a_2$ as optimal.

- The second two rows capture the interventional reward distribution, without the confounding effect of $\pi_b$. Delphic ORL provides an estimate of this, correctly identifying high delphic uncertainty in confounded states $a_0$ or $a_2$. The resulting policy chooses $a_1$ as the preferable action.

The ground-truth interventional reward (row 3) determines the expected reward for each action, showing that the delphic ORL policy ($a_1$) outperforms the offline RL one. Overall, this experiment further supports our method and helps understand its mechanisms.

## E.2  SEPSIS ENVIRONMENT

**Ablation Study: Delphic Uncertainty.**  In Figure 10a, we find that delphic uncertainty is highest on the sepsis dataset when treatment involves vasopressors. By design of the simulation (Oberst and Sontag, 2019), this treatment is the only one for which patient evolution is confounded by the hidden diabetic status, which further supports the conclusion that delphic uncertainty captures model

**Information and consent form**

*Evaluation of new reinforcement learning algorithm applied to treatment optimisation*

Participant (full name): .................................

Conducting person (full name):  REDACTED
Contact project team:  REDACTED
Ethics Commission Officer:  REDACTED

We would like to ask you if you are willing to participate in our research project. Your participation is voluntary. Please read the text below carefully and ask the conducting person about anything you do not understand or would like to know.

**What is investigated and how?**
We developed an algorithm to learn optimal treatment strategies and would like to validate it by measuring agreement with medical doctors.
To achieve this, we have designed a questionnaire of randomised synthetic patient charts with varying amount of information, each with a choice between two treatment options.
The anticipated benefit of this experiment is to facilitate the validation of a novel reinforcement learning algorithm, with possible application to treatment recommendation.

**Who can participate?**
Certified physicians, preferably with expertise in intensive care.

**What am I supposed to do as a participant?**
Please choose the treatment option you believe would lead to optimal outcomes after examining patient charts. You can choose not to answer if you feel you do not have enough information to make a decision.

**What are my rights during participation?**
Your participation in this study is voluntary. You may withdraw your participation at any time without specifying reasons and without any disadvantages.

**What risks and benefits can I expect?**
There are no known risks associated with participating in this study. All collected data will be entirely anonymous.
A potential benefit of participating is to facilitate the validation of a novel algorithm, which could be useful for future treatment recommendation systems.

**What data is collected from me and how is it used?**
Anonymised preference over treatment options for different patient charts will be collected. No personally identifying data will be collected and no third-party providers will handle any personal data.

We aim to publicly release anonymised preference results together with the paper and code to enable reproducibility. Anonymised preference data will also be stored on a secure computing infrastructure (REDACTED).

**Who funds this study?**
REDACTED

**Complaints office**
REDACTED

1

---

**Consent Form**

I, the participant, confirm by my signature that:

- I have read and understood the study information. My questions have been answered completely and to my satisfaction.

- I comply with the inclusion and exclusion criteria for participation described above. I am aware of the requirements and restrictions to be observed during the study.

- I have had enough time to decide about my participation.

- I participate in this study voluntarily and consent that my personal data be used as described above.

- I understand that I can stop participating at any moment.

I would like to be informed about the results of this work

☐ Yes, Name and Phone Number or Email: .............................
☐ No

Full name of participant

.................................

..........................................          ..........................................
Place, Date                                        Signature participant

..........................................          ..........................................
Place, Date                                        Signature conducting person

2

Figure 8: **Full text of information and consent letter given to participants.**

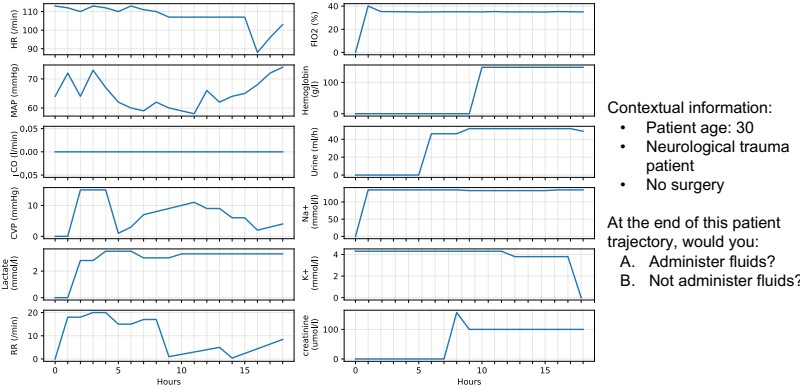

Figure 9: **Illustration of action ranking by medical experts.** Synthetic patient trajectories and a varying degree of contextual information (varying $|\mathcal{Z}|$ and $\Gamma$) are given to clinicians, who must rank the treatment options in terms of expected patient outcomes.

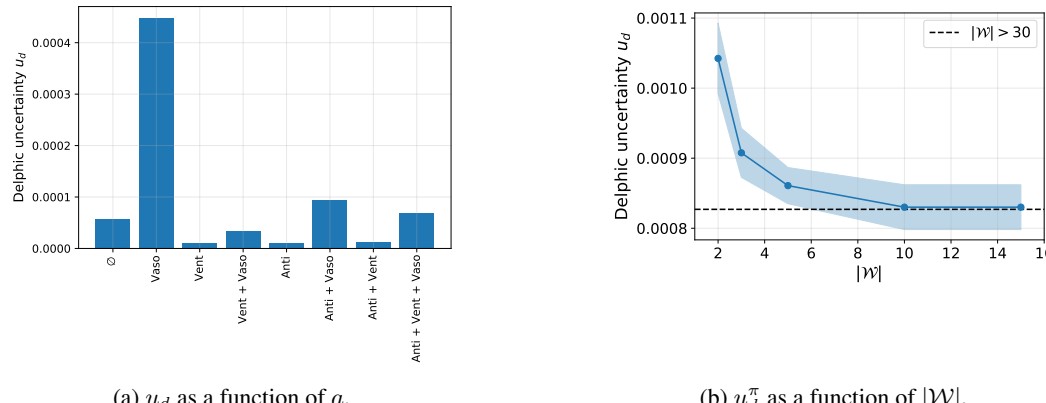

(a) $u_d$ as a function of $a$.
(b) $u_d^\pi$ as a function of $|\mathcal{W}|$.

Figure 10: **Ablation Study: Delphic Uncertainty.** (a) Delphic uncertainty is highest under vasopressors in the sepsis environment, correctly identifying their confounded effect (Abbreviations: Vaso = Vasopressors, Anti = Antibiotics, Vent = Ventilation). (b) Empirically, only a small number of compatible worlds (for sepsis, $|\mathcal{W}| \approx 10$) is necessary to obtain an asymptotic estimate of $u_d$.

bias due to hidden confounding. As a result, delphic ORL penalises more strongly the action of taking vasopressors only (which induces high uncertainty with respect to possible confounders) than other actions with a more certain value estimate, including combinations of vasopressors with other treatments. Naturally, the performance of the resulting policy is weaker than a baseline with access to $z$ in Figure 5, but is stronger than a policy based on a value function with hidden confounding biases.

In Figure 10b, we note that a only small number of world models (for sepsis, $|\mathcal{W}| \approx 10$) is necessary to obtain an estimate of delphic uncertainty consistent with a large number of world models. This motivates our practical choice to only consider a small set of world models to obtain a reasonable estimate of uncertainty for Delphic ORL, but warrants further theoretical work establishing guarantees and probability of correctness.

**Ablation Study: Delphic ORL.** In Figure 11, we study the performance of Delphic ORL as a function of hyperparameter $\gamma$, interpolating between a naive implementation of Offline RL for very low values of $\gamma$ (virtually no penalty) and an excessively pessimistic algorithm, where the confounding penalty overcomes any possible high-reward behaviour.

Next, Table 6 compares the performance of **different approaches to implement pessimism** with respect to delphic uncertainty. We find that our approach proposed in the main paper, based on penalising the target for the Bellman update, performs best in this experimental setting (sepsis dataset with $\Gamma = 46$). Weighting-based approaches also show promising performance (either matching or improving the performance of BC and CQL, respectively), which may be an avenue for further work

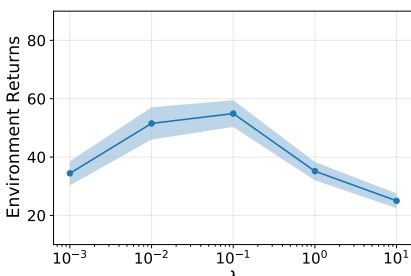

| Algorithm | Environment Returns |
|---|---|
| Online RL | $67.8 \pm 1.1$ |
| BC | $38.5 \pm 4.5$ |
| BCQ | $18.5 \pm 2.4$ |
| CQL | $31.1 \pm 3.5$ |
| CQL with history-dependent $Q$ | $26.9 \pm 3.1$ |
| CQL with history-dependent $Q, \pi$ | $23.5 \pm 2.0$ |
| Delphic ORL with $\Gamma = 46$ | $32.7 \pm 4.8$ |
| Delphic ORL (worst-case $Q_w^\pi$) | $18.1 \pm 2.0$ |
| Delphic ORL ($u_d$ Threshold) | $24.6 \pm 3.4$ |
| Delphic BC (Weighting) | $39.6 \pm 4.1$ |
| Delphic ORL (Weighting) | $44.7 \pm 4.2$ |
| Delphic ORL (Algo. 2) | $\mathbf{54.9} \pm 4.6$ |

Figure 11: **Performance results as a function of hyperparameter** $\lambda$ on the sepsis environment ($\Gamma = 46$).

Table 6: **Performance of different pessimism methods** on the sepsis environment ($\Gamma = 46$).

and fine-tuning. Modifying the Bellman target to only include actions below a certain uncertainty threshold was however found to be excessively pessimistic, and degraded performance compared to the base CQL algorithm. Model-based Offline RL and Delphic ORL were not included as their performance was never found to improve over a random baseline policy. Finally, setting the Bellman target as the lowest Q-function within the set of compatible worlds models is the closest setting to related work on robustness (Kallus and Zhou, 2018; Wang et al., 2021) and is also too pessmistic.

We compare to the sensitivity analysis framework for estimating the set of compatible worlds (Kallus and Zhou, 2018). We implement this through our variational approach, discarding compatible world models that do not satisfy the $\Gamma = 46$ assumption (all but 2 out of 10 trained compatible world models). This results in a poorer estimate of delphic uncertainty as the resulting delphic ORL performance is negatively affected (Delphic ORL with $\Gamma = 46$ in Table 6).

We hope this ablation study will motivate further work into the best possible approach to implement pessimism with respect to delphic uncertainty, to learn offline RL policies that are robust to hidden confounding bias.

We also measure whether **introducing history dependence** in training helps performance, as our algorithm indirectly uses history when computing delphic uncertainty (although both Q-function and policy are only state-dependent). In fact, we find that direct history dependence harms performance, which can be explained by the shift in history distribution between training (history distribution of the behavioural policy) and execution time (history distribution of the acting policy), as discussed in Ortega et al. (2021); Swamy et al. (2022). Overall, this result further motivates our focus on history-independent policies where confounding biases only affect the value function. In delphic ORL, only the delphic uncertainty term is used as penalty for $Q$, with limited possible utilisation of history as $u_d$ is computed over the entire dataset.

### E.3 REAL-WORLD CLINICAL DATASET

In this section, we provide additional evaluation metrics and investigations to understand the treatment strategies identified by the different algorithms considered, and in particular how Delphic ORL determines confounding-robust policies.

**Interpreting expert preferences.** Figure 7 shows the preference of clinicians for offline RL policies (Delphic ORL or CQL) over the BC policy. A score of 0% means clinicians always prefer the BC policy, whereas 100% means they always prefer the offline RL one. Overall, we found the preferences of the clinicians to be noisy (66% agreement in action preferences on average), which explains why preferences remain close to 50%. Under no confounding ($\mathcal{Z} = \emptyset$), clinicians prefer the BC policy to both offline RL ones (preference $<50\%$), as it should accurately capture doctors' decision-making in this non-confounded setting. Optimising a reward signal (necessarily distinct from doctors' internal objective) for offline RL therefore gives treatment policies that do not match the doctor's preference *as well*.

As confounding is increased by removing critical decision-making factors from the data ($\mathcal{Z} = \{\text{all}\}$), however, doctors prefer the delphic ORL policy to the BC one $\sim58\%$ of the time. This highlights both that BC degrades in performance due to missing information, and that delphic ORL becomes more favourable as it takes conservative actions in the face of unknown possible confounding (see

below). On the other hand, doctors now prefer the CQL policy to the BC one only $\sim$41% of the time. This suggests that CQL, picking up on spurious correlations, proposes an even worse treatment policy than BC with missing information.

**Understanding the delphic ORL policy.** Table 7 provides a quantitative analysis of the disparities in action choices between different algorithms and the doctors' policy. As expected, behaviour cloning exhibits the closest resemblance to the doctors' treatment policy, which aligns with the characteristics of observational datasets. However, our proposed method outperforms behaviour cloning in terms of learning a distinct policy that deviates from the doctors' actions. These findings highlight the unique capabilities of our method in capturing important features and patterns beyond the direct imitation of doctors, enabling the model to make informed decisions that may differ from the observational data and potentially lead to improved treatment outcomes.

Table 7: **Difference in action choices from** $\mathcal{D}_{test}$ across different algorithms (%). Our method learns a distinct policy from the doctors'. Mean and 95% CIs over 10 runs. Highest and overlapping values in bold.

| Confounders $\mathcal{Z}$ | BCQ | BC | CQL | **Delphic ORL** |
|---|---|---|---|---|
| All below | **32.2** $\pm$ 1.3 | 19.7 $\pm$ 1.1 | **33.4** $\pm$ 0.9 | **35.2** $\pm$ 1.5 |
| {age} | **31.5** $\pm$ 1.3 | 12.8 $\pm$ 0.4 | 27.3 $\pm$ 0.3 | **32.3** $\pm$ 0.5 |
| {neuro. diag.} | 31.1 $\pm$ 1.3 | 16.3 $\pm$ 1.0 | **34.3** $\pm$ 1.3 | 30.6 $\pm$ 1.1 |
| {gastro. diag.} | **27.1** $\pm$ 1.1 | 14.3 $\pm$ 0.9 | **28.9** $\pm$ 1.1 | **29.4** $\pm$ 1.3 |
| {trauma} | **30.1** $\pm$ 1.5 | 12.8 $\pm$ 0.4 | 24.2 $\pm$ 0.4 | 22.2 $\pm$ 0.7 |
| {cardio. diag.} | 28.7 $\pm$ 1.3 | 18.8 $\pm$ 1.2 | **36.2** $\pm$ 1.3 | 29.6 $\pm$ 1.6 |
| {endo. diag.} | **27.4** $\pm$ 1.3 | 13.5 $\pm$ 0.8 | **27.3** $\pm$ 0.9 | 23.1 $\pm$ 0.9 |
| {hemato. diag.} | **30.1** $\pm$ 1.5 | 12.4 $\pm$ 0.8 | 24.4 $\pm$ 1.1 | 23.6 $\pm$ 0.8 |
| {weight} | **28.9** $\pm$ 1.3 | 13.2 $\pm$ 0.4 | 25.4 $\pm$ 0.6 | 23.6 $\pm$ 1.2 |
| {sedation} | **30.5** $\pm$ 1.5 | 14.5 $\pm$ 0.7 | 25.1 $\pm$ 1.1 | 25.8 $\pm$ 1.0 |
| {resp. diag.} | 27.7 $\pm$ 1.3 | 14.2 $\pm$ 0.6 | 28.5 $\pm$ 1.1 | **25.2** $\pm$ 1.2 |
| {intoxication} | 25.7 $\pm$ 1.1 | 12.6 $\pm$ 0.6 | **26.3** $\pm$ 0.6 | 23.1 $\pm$ 0.9 |
| {surgical status} | **27.3** $\pm$ 1.3 | 14.3 $\pm$ 0.6 | 23.9 $\pm$ 0.8 | 22.1 $\pm$ 1.2 |
| {ortho. diag.} | **25.6** $\pm$ 1.1 | 12.3 $\pm$ 0.6 | 24.1 $\pm$ 1.1 | 22.3 $\pm$ 1.0 |
| {sepsis} | **26.1** $\pm$ 1.1 | 15.6 $\pm$ 0.8 | 23.5 $\pm$ 0.8 | 21.9 $\pm$ 1.2 |
| $\emptyset$ | **25.3** $\pm$ 0.9 | 12.2 $\pm$ 0.4 | 23.1 $\pm$ 0.8 | 21.7 $\pm$ 0.9 |

Following published recommendations on evaluating RL models in observational settings (Gottesman et al., 2018), we also analyse where policies differ most from the action choices in the observational dataset, and find that the policy learned by Delphic ORL diverges most at high disease severity (SOFA scores $\approx$ 15-20). In these cases, our policy appears to prescribe less fluids and vasopressors than in the data – which may be reasonable if unsure about possible adverse effects of an intervention. This relates to a comment received from one of the expert clinicians interviewed: "If I lack information about a patient [e.g. age, medical background and deliberately excluded variables], I would probably be more conservative with my treatment". Finally, we note a closer match to actions in the observational data at very high disease severity (SOFA score > 20), where negative rewards for *not* taking a therapeutic action outweighs potential confounding bias. Beyond this analysis, further insights could be gained by comparing interpretable representations of the trained policies (Pace et al., 2022), but we leave this as further work.

