# OpenReview forum: "Delphic Offline Reinforcement Learning under Nonidentifiable Hidden Confounding"
_ICLR.cc/2024/Conference — ICLR 2024 poster_

### Official Review · Reviewer_Lkaz · 2023-10-29

**Soundness:** 3 good
**Presentation:** 4 excellent
**Contribution:** 3 good
**Rating:** 8
**Confidence:** 3

**Summary:**

The paper addresses the problem of nonidentifiable confounding in RL. In this regard, a new notion of so-called delphic uncertainty is introduced in addition to aleatoric and epistemic uncertainties. An offline RL algorithm is proposed that penalizes taking actions with high delphic uncertainty. The performance is reported on both synthetic and real data.

**Strengths:**

- The presentation is excellent with adequate illustrations that helped my understanding.
- The new notion of uncertainty is insightful and important in the real application of RL.
- The algorithm and employed strategies sound reasonable to me.

**Weaknesses:**

I generally enjoyed reading this paper, but there are a few things that I wish were discussed in more depth or clarified:
1. I'm a little confused about how should I decide whether a world is compatible or not. Apparently, I can start with any confounder space dimensionality, prior $p(z)$ and model architectures and then estimate the parameters using ELBO, and then I have a compatible world? How far I can go here or what should I see to say a world is not compatible here?
2. The behavior policy is in general a context-aware policy. So, I'd expect enforcing similarity to the behavior policy might result in some sort of context awareness. Is this the case? For instance, in the illustrative bandit example, an optimal context-independent policy only explores $a_0$ and $a_2$, which in World 2 is very different from the behavioral policy and seems to be suboptimal compared to a uniform policy. So, if I get it right, the similarity to behavioral policy is encouraged in this setting with unobserved confounders.
3. As a similar question to the previous question, don't we expect avoiding actions $(s,a)$ with high delphic similarity to result in a policy more similar to the behavioral policy? It seems to be the opposite: page 8 "... we also studied the discrepancy of our trained policy with that in the data. Particularly, we compared the actions taken by our policy and the policy in the data and found that ... our policy was significantly different."
4. Could you elaborate on how counterfactual $Q_w^\pi$ is estimated using importance sampling in Section 5.1? Also, I'm not sure where in appendix C is referred to.
5. I'm not sure how to think about a reasonable $\Gamma$. Isn't Figure 7 concerning?

Recommendations:
I do not see the name of real data or the details explicitly mentioned in the main text. But I can see, as you have cited, Raghu et al. have pioneered using RL in the setting of sepsis treatment using a publicly available MIMIC dataset. If this is not your data, I recommend reporting performance on MIMIC for further reproducibility. Also, looking at the recent citations of Raghu et al., it seems CQL might not be the best baseline here. For instance, Shirali, Schubert, and Alaa have included medical guidelines as potential contexts for RL formulation, with better performance and higher similarity to behavioral policy. You may want to consider more recent works as a baseline or use their observations in favor of context awareness.

**Questions:**

Please refer to the Weaknesses. I'm happy to update my scores after hearing your thoughts.

---

> ### Author Response · Authors · 2023-11-17
> **Response to Reviewer Lkaz (1/3)**
>
> Dear Reviewer Lkaz,
>
> Many thanks for taking the time to read our work and for your positive feedback.
>
> ## Q1. Compatible worlds
> *"I'm a little confused about how should I decide whether a world is compatible or not. Apparently, I can start with any confounder space dimensionality, prior $p(z)$ and model architectures and then estimate the parameters using ELBO, and then I have a compatible world? How far I can go here or what should I see to say a world is not compatible here?"*
>
> Thank you for your question, which allows us to stress the different contributions of our work.
>
> **Formalising compatible worlds.** In our basic definition in Section 4, a compatible world is any model that *maximises the likelihood of the observational distribution*. In the limit of infinite data and no environment stochasticity (i.e., no epistemic or aleatoric uncertainty), finding a compatible world is equivalent to solving this optimisation problem.
>
> In the identifiable setting, all compatible world models should collapse to the true identifiable distribution of confounders and confounder-dependent policy and value function model. In the non-identifiable setting, our motivation is not to attempt to recover the confounders directly, but rather to obtain a plausible *set* of confounder models that fit the observational data.
>
> One of the main contributions of our work is therefore this formalism of compatible world models and of delphic uncertainty. This basic definition does not require any assumptions: any measure of the disagreement between compatible world models captures delphic uncertainty.
>
> **Modelling compatible worlds in practice.** A second major contribution is therefore one practical instantiation of delphic uncertainty quantification. Within many possible modelling choices, we use variational inference to estimate confounder distributions. This approach demonstrates the effectiveness of delphic offline RL in our experiments but leaves open further work in investigating alternative modelling approaches for compatible worlds (e.g. different priors, potentially based on prior knowledge such as the sensitivity analysis assumption). We investigate this empirically in our answer to Reviewers 3wVu and x1vz.
>
> We hope this answer addresses your confusion and look forward to hearing your thoughts in follow-up.
>
> ## Q2. Expected pessimistic behaviour
> *"The behavior policy is in general a context-aware policy. So, I'd expect enforcing similarity to the behavior policy might result in some sort of context awareness. Is this the case? For instance, in the illustrative bandit example, an optimal context-independent policy only explores $a_0$ and $a_2$, which in World 2 is very different from the behavioral policy and seems to be suboptimal compared to a uniform policy. So, if I get it right, the similarity to behavioral policy is encouraged in this setting with unobserved confounders."*
>
> This is a very interesting question that resonates with offline RL work, in which it was found that pessimism with respect to epistemic uncertainty, in practice, does result in regularization towards the behaviour policy (Fujimoto and Gu, 2021). Still, by exploiting the reward signal, offline RL methods result in a **different policy than that in the data**.
>
> The behaviour policy is only accessible in the data in its marginalised form equal to $E_{\nu(z)} [ \pi_b (a|s,z)]$ where the true confounder distribution $\nu(z)$ is nonidentifiable. While enforcing similarity to this marginal $\tilde{\pi}_b(a|s)$ should help minimise epistemic uncertainty (Fujimoto and Gu, 2021; Levine et al., 2020), we wonder how this would provide context-awareness. It is **possible** that the counterfactual $Q^{\tilde{\pi}_b}$ could be **less prone to delphic uncertainty** than the $Q^{\tilde{\pi}}$ of any other context-independent policy $\tilde{\pi}$. Still, as our algorithm is formulated to measure and penalise the uncertainty for the policy $\tilde{\pi}$ at hand, we obtain a different behaviour than simply regularising to $\tilde{\pi}_b$ (as implemented by our offline RL baselines). This would be an interesting theoretical investigation in further work, and we have made a note of this possible behaviour of the penalty in Section 5.2 of our revised manuscript.
>
> It is also an interesting observation that pessimism wrt delphic uncertainty in our bandit example could result in a policy that resembles the marginalised behaviour policy. We explicitly investigate this in Appendix E.1 and detect higher delphic uncertainty on actions prone to confounding, thus identifying the optimal context-independent policy as choosing $a_1$. In terms of probabilities, it is indeed **closer to the uniform marginal policy** $\tilde{\pi}_b(a|s)$ than the offline RL policy, but still **differs from** -- and performs better than -- $\tilde{\pi}_b$.

---

> > ### Author Response · Authors · 2023-11-17
> > **Response to Reviewer Lkaz (2/3)**
> >
> > ## Q3. Expected pessimistic behaviour
> >
> > *"As a similar question to the previous question, don't we expect avoiding actions $(s,a)$ with high delphic similarity to result in a policy more similar to the behavioral policy? It seems to be the opposite: page 8 "... we also studied the discrepancy of our trained policy with that in the data. Particularly, we compared the actions taken by our policy and the policy in the data and found that ... our policy was significantly different."*
> >
> > In Appendix E.3, Table 7, we quantify the difference in actions taken by learned policies and by doctors. As expected, behaviour cloning shows the closest resemblance to the doctors’ treatment policy, and Delphic ORL learns a **distinct** policy by exploiting the reward signal -- which justified our comment on page 8. Still, it shows **comparable or greater similarity to the doctors' actions than offline RL** baselines, which both already implement a form of regularisation towards the marginal behaviour policy. Note that Delphic ORL builds on top of CQL to only add the delphic uncertainty penalty.
> >
> > Overall, Delphic ORL does enforce, as you suggest, a form of similarity to the marginal behaviour policy $\tilde{\pi}_b(a|s)$, but in a way that *differs* from offline RL methods by weighting each state and action by our measure of possible confounding bias.
> >
> > Thank you again for these insightful questions. We have commented on the above in Section 6.2 in our revised manuscript and look forward to hearing your thoughts in follow-up.
> >
> > ## Q4. Counterfactual estimates of $Q_w$
> > *"Could you elaborate on how counterfactual $Q^{\pi}_w$ is estimated using importance sampling in Section 5.1? Also, I'm not sure where in appendix C is referred to."*
> >
> > Our approach is to change the policy term in $P_{\pi_b}$ to obtain counterfactual estimates, using the one-step importance sampling estimator
> > $$Q_{\theta_w}^{\pi} (s,a) =E_{\tau \sim \mathcal{D}}  E_{z \sim \nu_{\theta_w}(z|\tau)} [\frac{\pi(a|s)}{\pi_{b,\theta_w}(a|s,z)} Q_{\theta_w}^{\pi_b} (s,a,z) ],$$
> > using our estimate of the context-aware behavioural policy $\pi(a|s,z)$ (*local* to this world model and to its confounder distribution) as well as the latest policy estimate $\pi$.
> >
> > Delphic uncertainty builds upon the result that compatible worlds all appropriately model the observational distribution, but diverge in estimating counterfactual quantities -- that change one factor constituing $P_{\pi_b}$ in Equation (1). As we are motivated by RL, we choose to change the policy term, but any other could be considered. In Appendix C.1., page 20, we propose an alternative intervention on the world models to give a counterfactual quantity, this time by changing the local confounder distribution instead of the policy. While we did not investigate this quantitatively in depth, this forms a possible direction for further work.
> >
> > ## Q5. Scale of $\Gamma$ and Clarification of Figure 7
> > *"I'm not sure how to think about a reasonable $\Gamma$."*
> >
> > We adopt $\Gamma = \max_{s,a,z,z'} \frac{\pi_b(a|s,z)}{\pi_b(a|s,z')}$ as a measure of confounding strength for consistency with prior work (Kallus and Zhou, 2020; Jesson et al., 2021; Kallus et al., 2019; Namkoong et al., 2020). Its scale measures the **extent to which the behavioural policy depends on $z$ in choosing an action**, with $\Gamma=1$ for no dependence on confounders. In contrast to these works, $\Gamma$ is *not* a hyperparameter to be tuned in our approach. We merely use it as a descriptive characteristic of the environments we consider.
> >
> > When applied to measuring confounding in our simulated and real datasets, we find that our values of $\Gamma$ reach greater values than prior work such as Kallus and Zhou, 2020, whose experiments only go up to $\Gamma \approx 20$. This can be explained by the much higher complexity of the environments we consider (continous variables, over 200 state dimensions, 14 confounder dimensions -- resulting in a more sensitive measure). It is however encouraging that our approach retains performance even under these large values of $\Gamma$.
> >
> >
> > *"Isn't Figure 7 concerning?"*
> >
> > To clarify, Figure 7 shows the **preference** of clinicians for offline RL policies (Delphic ORL or CQL) over the BC policy: 0% means clinicians always prefer the BC policy, 100% means they always prefer the offline RL one. Overall, we found the **preferences of the clinicians to be noisy** (66% agreement on average), which explains why preferences between BC and offline RL policies remain **close to 50%**.
> >
> > Under no confounding ($\mathcal{Z}=\emptyset$), clinicians prefer the BC policy over both offline RL policies (preference <50%). BC should accurately capture doctors' decision-making in this non-confounded setting. Optimising a reward signal (necessarily distinct from doctors' internal objective) for offline RL therefore gives treatment policies that do not match the doctor's preference as well.
> >
> > (continued in next comment)

---

> > > ### Author Response · Authors · 2023-11-17
> > > **Response to Reviewer Lkaz (3/3)**
> > >
> > > (answer to Q5 continued)
> > > As confounding is increased by removing critical decision-making factors from the data ($\mathcal{Z}={all}$), however:
> > > * **Doctors prefer the delphic ORL policy to the BC one** ~58% of the time. This highlights both that BC degrades in performance due to missing information, and that delphic ORL becomes more favourable as it takes conservative actions in the face of unknown possible confounding. One of the clinicians noted: "If I lack information about a patient [e.g. age, medical background and deliberately excluded variables], I would probably be more conservative with my treatment" (see Appendix E.2).
> > > * On the other hand, doctors now prefer the CQL policy to the BC one only ~41% of the time. This suggests that CQL, picking up on spurious correlations, proposes an **even worse treatment policy** than BC with missing information.
> > >
> > > We hope that these clarifications address your concerns with respect to Figure 7, and have included these in Appendix E.3.
> > >
> > >
> > > ## Q5. Dataset details and baselines
> > > *"Recommendations: I do not see the name of real data or the details explicitly mentioned in the main text. But I can see, as you have cited, Raghu et al. have pioneered using RL in the setting of sepsis treatment using a publicly available MIMIC dataset. If this is not your data, I recommend reporting performance on MIMIC for further reproducibility."*
> > >
> > > Thank you for pointing this out, we have ensured to include the name of our dataset (HiRID, released by Hyland et al., 2020) as well as some details in the main text of our paper.
> > >
> > > Our motivation behind using HiRID over MIMIC is the **scale** of the former dataset, with many more variables monitored (over 200 against under 50 in the latter) at a much higher time resolution, including the existence of **diagnosis information** -- which we used as possible confounder values. HiRID is publicly available on the same platform as MIMIC (PhysioNet).
> > >
> > > *"Also, looking at the recent citations of Raghu et al., it seems CQL might not be the best baseline here. For instance, Shirali, Schubert, and Alaa have included medical guidelines as potential contexts for RL formulation, with better performance and higher similarity to behavioral policy. You may want to consider more recent works as a baseline or use their observations in favor of context awareness."*
> > >
> > > Thank you very much for pointing out this relevant related work, which we now include in our revised manuscript for context awareness.
> > >
> > > Note that the method proposed by Shirali et al., 2023, focuses on pruning actions in a high-dimensional action space before applying CQL. They find performance is highest when they can reduce the action space to 12 or 4 dimensions. As the environments we consider in this work are already only 8-dimensional, we conjecture that Pruned CQL would provide **marginal gains on top of CQL**.
> > >
> > > CQL remains an established offline RL baseline in related discrete actions-space applications (Kondrup et al., 2023; Shiranthika et al., 2022; Levine et al., 2020), and we demonstrate both its **limitations** in the context of hidden confounding, as well as how our simple but effective penalty **recovers good performance** in this setting. We believe this demonstrates the added value of our method, but look forward to hearing your thoughts in follow-up.
> > >
> > > We clarify this point in Section 6.2 of our revised manuscript.
> > >
> > > ---
> > > Thank you again for your feedback and questions which helped improve our paper. Thank you also for offering to increase your score -- we hope this will be the case if we have addressed your remaining concerns.
> > >
> > >
> > > ### References
> > >
> > > Kondrup et al. Towards Safe Mechanical Ventilation Treatment Using Deep Offline Reinforcement Learning. AAAI. 2023.
> > >
> > > Shiranthika et al. Supervised Optimal Chemotherapy Regimen Based on Offline Reinforcement Learning. IEEE Journal of Biomedical and Health Informatics. 2022.
> > >
> > > Levine et al. Offline RL: Tutorial, review, and perspectives on open problems. 2020.
> > >
> > > Fujimoto and Gu. A minimalist approach to offline reinforcement learning. NeurIPS, 2021.
> > >
> > > Kallus et al. Interval Estimation of Individual-Level Causal Effects Under Unobserved Confounding. AISTATS, 2019.
> > >
> > > Kallus and Zhou. Confounding-robust policy evaluation in infinite-horizon reinforcement learning. NeurIPS, 2020.
> > >
> > > Jesson et al. Quantifying ignorance in individual-level causal-effect estimates under hidden confounding. ICML, 2021.
> > >
> > > Namkoong et al. Off-policy policy evaluation for sequential decisions under unobserved confounding. NeurIPS, 2020.

---

> > > > ### Comment · Reviewer_Lkaz · 2023-11-20
> > > >
> > > > Thank you for your comprehensive response. I increased my score as I believe this paper is a major contribution, addressing an interesting topic with sufficient detail and support. Below are quick replies to some of the points you made:
> > > >
> > > > Q1. Thank you for clarifying the contributions. I agree formalizing compatible worlds is itself a contribution. On the modeling compatible worlds in practice, your choice made sense to me. Your response to other reviewers was convincing for me at this point. Still, I think alternative modeling approaches for compatible worlds will remain an outstanding challenge for future works.
> > > >
> > > > Q3. Table 7 and the clarification on page 8 were helpful for my understanding. This question is resolved.
> > > >
> > > > Q4. Your clarification was very helpful. I agree; it's interesting that, at some point, for very large $\Gamma$s, physicians start to prefer Delphic ORL. However, this happens when $\Gamma$ is too large. That's why I was thinking what $\Gamma$ is a large gamma and potentially unrealistic. Anyway, it's always tricky to evaluate BC vs. offline RL by asking physicians, as BC is supposed to mimic physicians, and it's natural that they find it promising. To me, the most interesting observation is how CQL and Delphic ORL diverge in two different directions.

---

> > > > > ### Author Response · Authors · 2023-11-23
> > > > >
> > > > > Dear Reviewer Lkaz,
> > > > >
> > > > > Many thanks for your positive feedback and for increasing your score. We are glad our response was useful and agree with your comments, which we will make sure to include in our next version of the manuscript. Thank you again for helping us improve our paper.

---

### Official Review · Reviewer_qrjW · 2023-10-30

**Soundness:** 4 excellent
**Presentation:** 3 good
**Contribution:** 3 good
**Rating:** 6
**Confidence:** 4

**Summary:**

The paper uses the contextual Markov Decision Process to model unobserved confounders in offline reinforcement learning. First, the authors define the class of contextual MDP that are compatible with the dataset. Then, based on a variance decomposition formula, the authors introduce the delphic uncertainty. Delphic uncertainty means the variance of policy performance across all compatible worlds. Then based on the delphic uncertainty term, the authors propose a penalty term for offline RL.

**Strengths:**

The method is applied to two real/semi-real-world medical datasets, which is very nice.

**Weaknesses:**

The probability setup of the paper is a bit unclear to me.

**Questions:**

1. On page 4, what does the notation $P_w \mapsto \Delta W$ mean? Do you mean $ P_w \in \Delta W$?

2. Right after Def 4.1, the author proposes to model the Q function as a random element "value model $Q_{\theta w}$ is defined by some stochastic model". Where is this randomness coming from? Isn't the value function of a policy just a deterministic function?

3. Related to the previous remark, the meaning of Theorem 4.2 is unclear to me. Could the authors detail the probability setup for this theorem? From my understanding, there is a measure over the space of compatible worlds (by the notation $E_w$), and then there is a measure over Q-value functions (by the notation $E_{\theta_w}$). What is the relationship between these measures?

---

> ### Author Response · Authors · 2023-11-17
> **Response to Reviewer qrjW**
>
> Dear Reviewer qrjW,
>
> Thank you very much for your feedback.
>
> ## Q1. Notation
> Thank you for pointing out the notation issue on $P_w$. We have corrected this in our manuscript.
>
> ## Q2. Value function stochasticity
> *"Right after Def 4.1, the author proposes to model the Q function as a random element "value model $Q_{\theta_w}$ is defined by some stochastic model". Where is this randomness coming from? Isn't the value function of a policy just a deterministic function?"*
>
> In Section 4, we consider $Q$ as a random variable capturing the sum of discounted rewards in a trajectory, $Q = \sum_{t=0}^{\infty} \gamma^t r(s_t, a_t, z)$, for a given $(s,a)$ pair acting as initial values. In the limit of infinite data and under no confounding, randomness in estimating this variable can be attributed to stochasticity in environment transition and reward functions or in the acting policy. This irreducible uncertainty is qualified as *aleatoric*.
>
> Limited data and hidden confounding introduce additional stochasticity in estimating $Q$. We formalise this in our next answer (Q3) where we clarify our probability setup. This approach mirrors that of prior work in offline RL (Yu et al., 2020), where stochasticity in the environment model or value function is both inherently modeled (measuring aleatoric uncertainty over $Q$) and quantified with respect to different data bootstraps (epistemic uncertainty).
>
> We look forward to hearing your thoughts on our answer.
>
>
> ## Q3. Probability setup
>
> *"Related to the previous remark, the meaning of Theorem 4.2 is unclear to me. Could the authors detail the probability setup for this theorem? From my understanding, there is a measure over the space of compatible worlds (by the notation $E_w$), and then there is a measure over Q-value functions (by the notation $E_{\theta_w}$). What is the relationship between these measures?"*
>
> Thank you for your question. We clarify our probability setup in the following.
> * Within the context of a world model $w$ and a parameterization of this world model $\theta_w$, we model a **probability distribution over episode returns**, $P(Q|\theta_w, w)$. For instance, this captures the environment stochasticity discussed above.
> * Next, the distribution of random variable $\theta_w$ depends itself on the specific world model $w$ it parameterises (i.e. the local confounder distribution being estimated, and how policy and environment depend on it). Our data and optimisation algorithm therefore define a distribution $P(\theta_w|w)$ **over the world model's parameter space**.
> * Finally, we posit a distribution $P(w)$ over the **compatible worlds** $\mathcal{W}$.
>
> Combining these measures, we obtain the following decomposition for the distribution of $Q$:
>
> $$P(Q) = \int \int P(Q | \theta_w, w) P(\theta_w | w) P(w) d\theta_w dw$$
>
> Note that in our work, $Q$ is subscripted with $\theta_w$ as a reminder for this dependence on $\theta$ and $w$. We have dropped this here for clarity.
>
> Within this framework, we denote the following expectations, using $x$ as a dummy random variable:
>
> $$E [x| \theta_w, w] = \int x P(x | \theta_w, w) dx$$
> $$E_{\theta_w} [x| w] = \int x P( \theta_w | w) d\theta_w$$
> $$E_{w} [x] = \int x P( w) dw$$
>
> For instance, this allows us to write:
> $$E [ Q ] = \int \int \int Q P(Q | \theta_w, w) P(\theta_w | w) P(w) dQ d\theta_w dw$$
> $$= E_{w} \left[E_{\theta_w} \left[E[Q | \theta_w, w] | w \right] \right]$$
>
> For variances, we define:
>
> $$Var (x| \theta_w, w) = \int x^2 P(x | \theta_w, w) dx - \left( \int x P(x | \theta_w, w) dx \right) ^2$$
> $$Var_{\theta_w} (x| w) = \int x^2 P( \theta_w | w) d\theta_w - \left( \int x P( \theta_w | w) d\theta_w \right) ^2$$
> $$Var_{w} (x) = \int x^2 P( w) dw - \left( \int x P( w) dw \right) ^2$$
>
> Following the proof in Appendix B, we obtain the variance decomposition in Theorem 4.2:
> $$Var(Q)= E_{w} \left[E_{\theta_w} \left[ Var(Q | \theta_w, w) | w \right] + Var_{\theta_w}\left(E[Q |  \theta_w, w] | w\right) \right] + Var_{w} (E_{\theta_w}[\mathbb{E}[Q| \theta_w, w]|w ])$$
>
>
> We hope this helps clarify our probability setup and have modified our manuscript to reflect this.
>
> ---
> Thank you again for your feedback and questions which helped improve our paper. We would be very grateful if you would increase your score if we have addressed your remaining concerns.
>
>
> ### References
>
> Yu et al. Mopo: Model-based offline policy optimization. NeurIPS, 2020

---

> > ### Comment · Reviewer_qrjW · 2023-12-01
> > **Thank you**
> >
> > I want to thank the authors for their clarifications. I increase the score accordingly.

---

### Official Review · Reviewer_x1vz · 2023-10-31

**Soundness:** 3 good
**Presentation:** 3 good
**Contribution:** 3 good
**Rating:** 8
**Confidence:** 3

**Summary:**

This paper studies the problem of offline learning an optimal independent policy. To achieve this goal, the authors propose the Delphic Offline RL algorithm that:
1. identifies the compatible world model for confounded MDP and learn the world-dependent value function $Q_w^\pi$;
2. incorporates pessimistic policy optimization using the estimated delphic uncertainty;
The experimental results show that Delphic ORL method achieves better performance under large confounding strength.

**Strengths:**

The paper is well motivated and the author proposed an interesting idea to incorporate the delphic uncertainty in pessimistic offline policy optimization. The writing is generally clear and easy to follow.

**Weaknesses:**

1. It seems that the estimator $\mathbb{Var}_w (Q^π_w (s, a))$ is not an unbiased estimator for the delphic uncertainty as the other two forms of uncertainty can still enter the estimation (noting that $Q_w^\pi$ is not the conditional expectation given by Theorem 4.2). I didn't see the author making effort to justify this point.

2. Is it necessary to evaluate the delphic uncertainty on a state-action level rather than evaluating the same thing for the total reward of the whole trajectory under  policy $\pi$? I'm not convinced of the soundness of the method here, as there might be some correlations between different state-action pairs in the Q function.

**Questions:**

It is assumed that the offline policy is known. What if $\pi_b$ is unavailable so that the importance sampling method has biased in estimating the value function? Is it possible to incorporate other unbiased estimation method in confounded POMDP setting like (Shi. et al, 2022)?


### Reference:

Shi, Chengchun, et al. "A minimax learning approach to off-policy evaluation in confounded partially observable markov decision processes." International Conference on Machine Learning. PMLR, 2022.

---

> ### Author Response · Authors · 2023-11-17
> **Response to Reviewer x1vz (1/2)**
>
> Dear Reviewer x1vz,
>
> Many thanks for your feedback.
>
> ## Q1. Practical definition of delphic uncertainty
> *"It seems that the estimator $Var_{w} (Q_{w}^{\pi} (s,a))$ is not an unbiased estimator for the delphic uncertainty as the other two forms of uncertainty can still enter the estimation (noting that $Q^{\pi}_{w}$ is not the conditional expectation given by Theorem 4.2)."*
>
> Thank you for pointing out this possible source of confusion.
>
> From our practical estimator of delphic uncertainty in Section 5.1 ($Var_{w} (Q_{w}^{\pi} (s,a))$), we recover its definition in Equation 1 ($Var_{w} (E_{\theta_w}[E [Q_{\theta_w}^{\pi} |  \theta_w, w]]$) iff $Q_{w}^{\pi} (s,a) = E_{\theta_w}[ E [Q_{\theta_w}^{\pi} |  \theta_w, w]| w ]$.
>
> * In practice, we achieve this by implementing each world model $w$ as an ensemble, with each particle parameterised by a value of $\theta_w$. We approximate the outer expectation over $\theta_w$ as follows:
> $$Q_{w}^{\pi} (s,a)= \frac{1}{m}\sum_{\theta_w} \mathbb{E} [ Q^{\pi}_{\theta_w}(s,a) | \theta_w, w ]$$
> where $m$ is the number of particles in the world model ensemble. This sample mean is an unbiased estimator of the true expectation.
>
> * Next, the inner expectation can be obtained by implementing each ensemble particle $Q^{\pi}_{\theta_w}(s,a) | \theta_w, w$ as a probabilistic model. We model the distribution of $Q^{\pi_b}$ (details in Appendix C.1) and use its mean to estimate $Q^{\pi}$ using importance sampling and marginalising over $z$.
>
> We hope this decomposition clarifies how our practical implementation of delphic uncertainty approximates our theoretical definition in Section 4. These uncertainty computation details are discussed in Appendix C.1 (page 19, now in bold), but we have modified Section 5.1 to reflect this better.
>
> Finally, note that both **pessimism with respect to *both* delphic and epistemic uncertainty** is generally desirable in the confounded offline RL setting we consider. As a result, while our separation of uncertainties proves useful to theoretically and empirically validate our proposed approach, retaining a biased estimate of both uncertainties (e.g. to avoid modelling ensembles of $\theta_w$) is still a useful quantity to penalise.
>
> ## Q2. Delphic uncertainty on a trajectory-level
>
> *"Is it necessary to evaluate the delphic uncertainty on a state-action level rather than evaluating the same thing for the total reward of the whole trajectory under policy $\pi$ ?"*
>
> This is an interesting question, as we could certainly compute the variance over compatible worlds for the return of a trajectory. Our reasons for focusing on the state-action level are two-fold:
> * First, focusing on $(s,a)$ pairs allows us to leverage the **MDP structure** of our problem, and thus to pool together delphic uncertainty estimates for similar pairs in different trajectories.
> * Next, and perhaps even more importantly, estimating the uncertainty at a state-action level allows us to use this in a **practical pessimistic RL algorithm**, in which we penalize $(s,a)$ pairs with high uncertainty as in prior work (Yu et al., 2020). Leveraging uncertainty over trajectories could also be conceivable (e.g. re-weighting offline trajectories based on their cumulative delphic uncertainty), but we are not aware of state-of-the-art offline RL algorithms that take this approach.
>
>
> *"I'm not convinced of the soundness of the method here, as there might be some correlations between different state-action pairs in the Q function."*
>
> Yu et al., 2020, also propose a method of estimating uncertainty at the state-action level (over the transition function in their model-based setting) and prove that penalising the value in such a way results in a provably pessimistic behaviour (see their Theorem 4.4) and a practical, effective algorithm. Correlations between state-action pairs do not pose a problem in their framework, but we are curious to discuss more if this remains a concern.
>
>
> ## Q3. Identifiability of the behavioural policy
> *"It is assumed that the offline policy is known. What if  $\pi_b$ is unavailable so that the importance sampling method has biased in estimating the value function?"*
>
> To clarify, our work does not assume that $\pi_b$ is known or identifiable. We learn it in the context of the local confounder distribution $\nu(z)$ in each world model. If $\pi_b$ were known, confounding would not be a source of error, and our problem setting would correspond to offline RL with partial-observability. In this setting (unlike the confounded one), $z$ could be identified from its effect on transitions or rewards -- up to epistemic and aleatoric uncertainty.

---

> ### Author Response · Authors · 2023-11-17
> **Response to Reviewer x1vz (2/2)**
>
> ## Q4. Partial identification methods
> *"Is it possible to incorporate other unbiased estimation method in confounded POMDP setting like (Shi. et al, 2022)?"*
>
> Thank you for providing this additional reference, we will ensure to cite it as an additional related work.
>
> The challenge of nonidentifiable confounding bias in deep offline RL is paramount (Gottesman et al., 2018; Tennenholtz et al., 2022), but remained unaddressed until now. Prior works consider the challenge in OPE (Kallus and Zhou, 2020; Shi et al., 2022) or online RL (Lu et al., 2018; Zhang and Bareinboim, 2019), but our work is the *first* to propose a practical solution to hidden confounding bias in offline RL. Our work builds on two key contributions:
> 1. Defining compatible worlds, i.e. that there exists a set of models that maximise the likelihood of the data but may disagree over counterfactual quantities. We propose one (out of many possible) implementation to find this set, but this is not the focus of our work. In fact, as noted in Table 1, any **other partial identification assumption** (e.g. known $\Gamma$ in sensitivity analysis) **could be adopted** to achieve this. We now also emphasize this point in Section 5.1.
> 2. Next, and more importantly, we propose a practical offline RL algorithm to mitigate this challenge. Rather than optimising for worst-case returns and obtaining excessively pessimistic behaviour (as is done in prior work, e.g. Shi et al., 2022, Wang et al., 2020; Kallus and Zhou, 2019), we propose to **measure uncertainty and to apply a pessimism penalty**. This is motivated by the theoretical guarantees and empirical success of offline RL algorithms that follow this approach (Levine et al., 2020; Yu et al., 2020).
>
> From this perspective, we agree that the following baselines are useful to measure the added value of our approach. We present our results below, for the sepsis environment with $\Gamma=46$, and include this in Appendix E.2, page 28.
> 1. comparing to the sensitivity analysis framework for estimating the set of compatible worlds (Kallus and Zhou, 2019). We implement this through our variational approach, discarding world models that do not satisfy the $\Gamma$ assumption (all but 2 out of 10 trained compatible world models). This results in a poorer estimate of delphic uncertainty as the resulting delphic ORL performance is negatively affected. [Updated Nov 19]
>
> 2. comparing to worst-case value optimisation (Wang et al., 2020; Kallus and Zhou, 2019).  As expected, optimising for the worst-case value function is excessively pessimistic:
>
> | Compatible worlds | Offline RL Algorithm | Environment Returns |
> |---|---|---|
> | Variational (ours) | Pessimism  (ours) | 54.9 $\pm$ 4.7 |
> | Variational  | Worst-case (Wang et al., 2020) | 18.1 $\pm$ 2.0  |
> | Sensitivity analysis bound (Kallus and Zhou, 2019) | Pessimism      | 32.7 $\pm$ 4.8      |
> | Sensitivity analysis bound (Kallus and Zhou, 2019) | Worst-case (Wang et al., 2020) | 18.6 $\pm$ 2.3      |
> ---
>
> Thank you again for your feedback and questions which helped improve our paper. We would be very grateful if you would increase your score if we have addressed your remaining concerns.
>
> ### References
>
> Gottesman et al. Evaluating reinforcement learning algorithms in observational health settings. arXiv preprint arXiv:1805.12298, 2018.
>
> Kallus and Zhou. Confounding-robust policy evaluation in infinite-horizon reinforcement learning. NeurIPS, 2020.
>
> Levine et al. Offline RL: Tutorial, review, and perspectives on open problems. 2020.
>
> Lu et al. Deconfounding reinforcement learning in observational settings. arXiv:1812.10576, 2018.
>
> Tennenholtz, et al. On covariate shift of latent confounders in imitation and reinforcement learning. ICLR, 2022.
>
> Wang et al. Provably efficient causal reinforcement learning with confounded observational data. NeurIPS, 2021.
>
> Yu et al. Mopo: Model-based offline policy optimization. NeurIPS, 2020
>
> Zhang, J. and Bareinboim, E. Near-optimal reinforcement learning in dynamic treatment regimes. NeurIPS, 2019.

---

> > ### Author Response · Authors · 2023-11-19
> > **Additional results in our rebuttal answer**
> >
> > Dear Reviewer x1vz,
> >
> > We have updated our response above to include results comparing to the sensitivity analysis framework for estimating the set of compatible worlds (Kallus and Zhou, 2019). We look forward to hearing your thoughts on our response and hope you will consider increasing your score if your concerns have been addressed.
> >
> > Many thanks.

---

> > > ### Comment · Reviewer_x1vz · 2023-11-22
> > >
> > > Thanks to the authors for clarifying their methods on estimation of the delphic uncertainty and incorporating pessimism in a practical way. Based on the authors' response, I believe that the authors can make a better argument on these points in the revision and perhaps unify the penalties for different sources of uncertainty in the algorithm (while I understand that the delphic uncertainty is still the highlight). I have increased my score accordingly.

---

> > > > ### Author Response · Authors · 2023-11-23
> > > >
> > > > Dear Reviewer x1vz,
> > > >
> > > > Thank you very much for your positive feedback and for increasing your score. We agree with your comment and are working on incorporating these suggestions into our next version of the manuscript. Thank you again for helping us improve our paper.

---

### Official Review · Reviewer_3wVu · 2023-11-01

**Soundness:** 3 good
**Presentation:** 3 good
**Contribution:** 4 excellent
**Rating:** 8
**Confidence:** 4

**Summary:**

* The authors address the unobserved confounding problem in offline reinforcement learning with pessimism over possible "worlds" (confounder values) compatible with the observation (distribution of trajectory).
* They define a new type of uncertainty "Delphic uncertainty" as the variance of Q value over the compatible (thus unidentifiable) worlds with theoretical decomposition with other types of uncertainties.
* Simulated evaluation and evaluation by experts clearly indicated that their method outperformed existing methods that do not address the Delphic uncertainty such as CQL and BC when strongly confounded.

**Strengths:**

* The unobserved confounding is a major issue in offline reinforcement learning.
* They investigate a minimal problem setting (contextual MDP) to reproduce it and propose a simple and intuitive method that models the uncertainty related to the confounding.
* The theory that decomposes the variance into several types of uncertainties motivates the approach well.
* Empirical evidence including evaluation by experts clearly supports their claim.

**Weaknesses:**

1. Baselines and environments tested are relatively limited (see also Question 1 and 2).
1. Not being a major concern, it would be more intuitively superior if an end-to-end formulation was possible, as in the CQL. The proposed method is divided into a step of learning multiple possible worlds and a step of pessimism using them.

**Questions:**

1. Intuitively, it seems that estimating $z$ from the trajectory and using it as $\pi(a|s,z)$ as in POMDP methods would improve performance for later steps $t$, but is such an extension possible? Also, is the proposed method still superior when such a POMDP method is used as a baseline? I'm wondering if the online identification of the world is possible within an episode through such a formulation.
1. The existing approaches for a similar setting are discussed (e.g. using partial identification) but not compared. Isn't it possible to compare them?
1. Is the $\max$ taken w.r.t. not only $z,z'$ but also $s,a$? If not, how $\Gamma(s,a)$ is summarized for an environment?

---

> ### Author Response · Authors · 2023-11-17
> **Response to Reviewer 3wVu (1/2)**
>
> Dear Reviewer 3wVu,
>
> Many thanks for taking the time to read our work and for your positive feedback.
>
> ## Q1. Leveraging confounder estimation at inference time
> *"Intuitively, it seems that estimating $z$ from the trajectory and using it as $\pi(a|s,z)$ as in POMDP methods would improve performance for later steps $t$, but is such an extension possible? Also, is the proposed method still superior when such a POMDP method is used as a baseline? I'm wondering if the online identification of the world is possible within an episode through such a formulation."*
>
> Thank you for this great question, which really highlights the challenge of our problem setting.
>
> In Appendix E.2, we investigate whether estimating and using confounder values $z$ at inference time helps improve the performance of our main baseline CQL. We implement this by allowing both $\{Q,\pi\}$ models to access the history of trajectories (from which $z$ can be estimated). We reproduce our results from Table 6 below: we find that **history-dependence actually degrades performance**.
>
> | Algorithm  | Environment Returns |
> |---|---|
> | BC | 38.5 $\pm$ 4.5      |
> | BCQ   | 18.4 $\pm$ 2.6      |
> | CQL   | 31.1 $\pm$ 3.3      |
> | CQL with history-dependent $Q$      | 26.9 $\pm$ 3.1      |
> | CQL with history-dependent $Q, \pi$ | 23.5 $\pm$ 2.0      |
> | Delphic Offline RL | 54.9 $\pm$ 4.7      |
>
> This result can be explained as follows. In partially-observed offline settings, incorporating a learned estimate for the confounders $z$ based on the history can be even more prone to confounding, as the identification model is trained on the history distribution of the *behavioural policy* and not that of the inference policy. This leads to a compounding of distribution shift at inference time, showing "latching behaviour" in imitation (Swamy et al., 2022) and "model delusion" in offline RL (Ortega et al., 2021). For this reason, in our approach to the problem, we chose to focus on history-independent policies where confounding biases only affect the value function, which we directly address through pessimism.
>
> ## Q2. Partial identification baselines
>
> *"The existing approaches for a similar setting are discussed (e.g. using partial identification) but not compared. Isn't it possible to compare them?"*
>
> The challenge of nonidentifiable confounding bias in deep offline RL is paramount (Gottesman et al., 2018; Tennenholtz et al., 2022), but remained unaddressed until now. Prior works consider the challenge in OPE (Kallus and Zhou, 2020), online RL (Lu et al., 2018; Zhang and Bareinboim, 2019), or using proxy variables (Wang et al., 2021), but our work is the *first* to propose a practical solution to hidden confounding bias in offline RL.
>
> Our work builds on two key contributions:
> 1. Defining compatible worlds, the set of models that maximise the likelihood of the data but disagree over counterfactual quantities. We propose one (out of many possible) implementation to find this set, but this is not the focus of our work. In fact, as noted in Table 1, any **other partial identification assumption** (e.g. known $\Gamma$ in sensitivity analysis) **could be adopted** to achieve this. We now also emphasize this point in Section 5.1.
> 2. Next, and more importantly, we propose a practical offline RL algorithm to mitigate this challenge. Rather than optimising for worst-case returns and obtaining excessively pessimistic behaviour (as is done in prior work, e.g. Wang et al., 2020; Kallus and Zhou, 2019), we propose to **measure uncertainty and to apply a pessimism penalty**. This is motivated by the theoretical guarantees and empirical success of offline RL algorithms that follow this approach (Levine et al., 2020; Yu et al., 2020).
>
> From this perspective, we agree that the following baselines are useful to measure the added value of our approach. We present our results below, for the sepsis environment with $\Gamma=46$, and include this in Appendix E.2, page 28.
> 1. comparing to the sensitivity analysis framework for estimating the set of compatible worlds (Kallus and Zhou, 2019). We implement this through our variational approach, discarding world models that do not satisfy the $\Gamma$ assumption (all but 2 out of 10 trained compatible world models). This results in a poorer estimate of delphic uncertainty as the resulting delphic ORL performance is negatively affected. [Updated Nov 19]
>
> 2. comparing to worst-case value optimisation (Wang et al., 2020; Kallus and Zhou, 2019).  As expected, optimising for the worst-case value function is excessively pessimistic:
>
> | Compatible worlds | Offline RL Algorithm | Environment Returns |
> |---|---|---|
> | Variational (ours) | Pessimism  (ours) | 54.9 $\pm$ 4.7 |
> | Variational  | Worst-case (Wang et al., 2020) | 18.1 $\pm$ 2.0  |
> | Sensitivity analysis bound (Kallus and Zhou, 2019) | Pessimism      | 32.7 $\pm$ 4.8      |
> | Sensitivity analysis bound (Kallus and Zhou, 2019) | Worst-case (Wang et al., 2020) | 18.6 $\pm$ 2.3      |

---

> > ### Author Response · Authors · 2023-11-17
> > **Response to Reviewer 3wVu (2/2)**
> >
> > ## Q3. Notation
> >
> > Thank you for pointing out this missing detail in our definition of confounding strength $\Gamma$. We have corrected this in our revised manuscript.
> >
> > ---
> >
> > Thank you again for your positive feedback and questions which helped improve our paper.
> >
> >
> > ### References
> >
> > Gottesman et al. Evaluating reinforcement learning algorithms in observational health settings. arXiv preprint arXiv:1805.12298, 2018.
> >
> > Kallus and Zhou. Confounding-robust policy evaluation in infinite-horizon reinforcement learning. NeurIPS, 2020.
> >
> > Levine et al. Offline RL: Tutorial, review, and perspectives on open problems. 2020.
> >
> > Lu et al. Deconfounding reinforcement learning in observational settings. arXiv:1812.10576, 2018.
> >
> > Ortega et al. Shaking the foundations: delusions in sequence models for interaction and control. arXiv:2110.10819, 2021.
> >
> > Swamy et al. Sequence model imitation learning with unobserved contexts. arXiv:2208.02225, 2022.
> >
> > Tennenholtz, et al. On covariate shift of latent confounders in imitation and reinforcement learning. ICLR, 2022.
> >
> > Wang et al. Provably efficient causal reinforcement learning with confounded observational data. NeurIPS, 2021.
> >
> > Yu et al. Mopo: Model-based offline policy optimization. NeurIPS, 2020
> >
> > Zhang, J. and Bareinboim, E. Near-optimal reinforcement learning in dynamic treatment regimes. NeurIPS, 2019.

---

> > > ### Author Response · Authors · 2023-11-19
> > > **Additional results in our rebuttal answer**
> > >
> > > Dear Reviewer 3wVu,
> > >
> > > Thank you again for your positive feedback.
> > >
> > > We have updated our response to include our comparison to the sensitivity analysis framework for estimating the set of compatible worlds (Kallus and Zhou, 2019). We look forward to hearing your thoughts on our response and hope to have addressed any remaining concerns.

---

> > > > ### Comment · Reviewer_3wVu · 2023-11-21
> > > >
> > > > Thank you for the additional results and discussion.
> > > > For Q2, the contribution would be clearer in a discussion comparing such methods that have similar concepts.
> > > > For Q1, a fully history-dependent policy may be too complex, but it may be conceivable to design a (semi-)online policy that schedules the degree of optimism/pessimism to identify the world and behaves optimistically while $t$ is small.
> > > > Anyway, based on the additional experimental results, I am generally satisfied with the authors' answers and I would like to keep this score.

---

> > > > > ### Author Response · Authors · 2023-11-23
> > > > >
> > > > > Dear Reviewer 3wVu,
> > > > >
> > > > > Many thanks for your positive feedback and for helping us improve our paper. We are working on including this discussion in our next version of the manuscript and will comment on this promising possible approach to incorporating partial identification.

---

### Author Response · Authors · 2023-11-17
**Response to all Reviewers**

Dear Reviewers,


Many thanks for taking the time to read our work and for your feedback which helped improve our manuscript.

All reviewers gave excellent soundness, presentation and contribution ratings. Reviewers agree that our method is well-motivated (3wVu, x1vz, Lkaz), and describe it as “simple and intuitive” (3wVu), “interesting” (x1vz), and “insightful” (Lkaz). Their feedback praised the quality of our presentation (x1vz, Lkaz) and the depth of our empirical analysis (3wVu, qrjW) and stressed the importance of the problem being tackled (3wVu, Lkaz).


We addressed reviewers’ comments and questions individually. We summarise some major points below:

* We emphasize that our work is the **first** to propose a **practical solution to hidden confounding bias in offline RL**.
    * Related approaches optimise for worst-case returns: we now show in Appendix E.2 that this leads to poor empirical performance (Reviewers 3wVu, x1vz).
* We show that incorporating partial identification or **history-dependence within the inference policy** can amplify confounding bias (Reviewer 3wVu).
* In Section 5.1, we stress that **any partial identification assumption** (e.g. sensitivity analysis) could be used to model compatible worlds (Reviewers 3wVu, x1vz, Lkaz).
    * [Updated Nov 19:] We show in Appendix E.2 that the sensitivity assumption results in lower offline RL performance than our proposed variational setup. We link to this result in Section 6.1 of our revised manuscript.
* In Sections 5.2 and 6.2, we comment on the possible behaviour of delphic offline RL in acting as a form of **regularisation towards the observed policy**, as in prior offline RL work (Reviewer Lkaz).
* We motivate our choice of dataset and comment on relevant **RL approaches for clinical data** in Section 6.2 (Reviewer Lkaz).
* We clarify our probability setup in Section 4 (Reviewer qrjW) and our practical estimation of delphic uncertainty in Section 5.1 (Reviewer x1vz).

Changes in our revised manuscript are highlighted in blue.

We look forward to hearing your follow-up thoughts. We would be very grateful if you would increase your scores if we have addressed your remaining concerns.

---

### Meta-Review · Area_Chair_tSPs · 2023-12-05

**Metareview:**

This paper addresses the issue of nonidentifiable hidden confounding in offline RL.  The delphic uncertainty is introduced to account for hidden confounding bias, using variation over world models compatible with the observations.  Practical estimation of delphic uncertainty is derived, based on which a pessimistic offline RL method is developed.  Experimental results show good promise of the method on simulation and real-world benchmarks with confounded data.

The paper addresses an important problem, and the solution is novel. The experimental results show it is effective. All reviewers and myself find the paper a good addition to the conference.

**Justification For Why Not Higher Score:**

The experiments are both based on medical data.  More breadth in application will be helpful.

**Justification For Why Not Lower Score:**

The paper addresses an important problem, and the solution is novel. The experimental results show it is effective. All reviewers and myself find the paper a good addition to the conference.

---

### Decision · Program_Chairs · 2024-01-16

Accept (poster)